# Discrete Latent Plans via Semantic Skill Abstractions

**Haobin Jiang**[1], **Jiangxing Wang**[1], **Zongqing Lu**[1,2*]
[1]School of Computer Science, Peking University
[2]Beijing Academy of Artificial Intelligence

## Abstract

Skill learning from language instructions is a critical challenge in developing intelligent agents that can generalize across diverse tasks and follow complex human instructions. Hierarchical methods address this by decomposing the learning problem into multiple levels, where the high-level and low-level policies are mediated through a latent plan space. Effective modeling and learning of this latent plan space are key to enabling robust and interpretable skill learning. In this paper, we introduce LADS, a hierarchical approach that learns language-conditioned discrete latent plans through semantic skill abstractions. Our method decouples the learning of the latent plan space from the language-conditioned high-level policy to improve training stability. First, we incorporate a trajectory encoder to learn a discrete latent space with the low-level policy, regularized by language instructions. Next, we model the high-level policy as a categorical distribution over these discrete latent plans to capture the multi-modality of the dataset. Through experiments in simulated control environments, we demonstrate that LADS outperforms state-of-the-art methods in both skill learning and compositional generalization. The code is available at https://github.com/PKU-RL/LADS.

## 1 Introduction

Creating an agent capable of understanding and executing natural language instructions has been a long-standing goal in both reinforcement learning (RL) and imitation learning (IL) (Luketina et al., 2019; Nair et al., 2022). This capability is essential for developing a generalist artificial intelligence (AI) that can follow human commands to perform a wide range of control tasks, such as playing virtual video games (Lifshitz et al., 2023) or performing real robotic manipulation (Brohan et al., 2022). Other generalist policies often condition on goal images (Nair et al., 2018) or states (Andrychowicz et al., 2017), where the goals are naturally grounded in the observation space. In contrast, language-conditioned policies face the challenge of grounding language into the observation space (Nair et al., 2022). With the development of vision-language models (VLMs), recent work has explored using pretrained models to achieve language grounding (Shridhar et al., 2022; Jiang & Lu, 2024). However, these methods typically focus on visual inputs and object grounding, with limited effectiveness in understanding numerical states, such as robotic proprioception, and motion information.

Learning a hierarchical language-conditioned policy is an effective approach to addressing the challenge of grounding language without being constrained by data modality. Hierarchical policy learning provides an intermediate representation that aligns language instructions and low-level control in a shared latent space. This approach significantly simplifies the complexity of language grounding by only requiring the language-conditioned *high-level policy* to map language instructions into a temporally and semantically abstract *latent plan* space (Lynch et al., 2020), rather than directly controlling actions. This process is often described as decomposing a task into multiple smaller *sub-tasks* (Rosete-Beas et al., 2023). The low-level policy is then responsible for generating the precise actions to interact with the environment. Specifically, it conditions on a latent plan vector and acts as a *skill* for completing the sub-task indicated by this latent plan. In addition, hierarchical policies offer the benefits of sample efficiency in learning complex, long-horizon tasks and improve generalization to unseen scenarios through task decomposition (Garg et al., 2022; Mees et al., 2022a).

---

*Correspondence to Zongqing Lu <zongqing.lu@pku.edu.cn>.

Recent work has advanced in this direction by implementing a hierarchical policy, where the low-level policy is learned from an offline dataset annotated with language instructions (Garg et al., 2022; Ju et al., 2024; Liang et al., 2024; Fu et al., 2024). The challenge in acquiring skill abstractions, including the low-level policy and the latent plan space, from language instructions lies in learning them in an unsupervised manner while ensuring the skills are both composable for complex, long-horizon tasks and interpretable for humans (Garg et al., 2022). To address this, these works opt for a *discrete* latent plan space for skill learning, as it offers better controllability and interpretability compared to continuous representations. Furthermore, discrete latent representations have also been proven effective in various fields such as world models (Hafner et al., 2020; 2023), image generation (Esser et al., 2021; Rombach et al., 2022), and audio codecs (Zeghidour et al., 2021).

While promise has been shown, there is a limitation in these methods where the high-level policy and low-level policy are trained jointly in an end-to-end manner (Garg et al., 2022; Ju et al., 2024; Liang et al., 2024). This can lead to potential training instability and difficulty, as the learning of the latent plan space and the language-conditioned high-level policy are entangled. The two components may affect each other's learning progress, resulting in index collapse in the codebook and thus requiring additional techniques to refine and stabilize the codebook (Ju et al., 2024). Inspired by task-agnostic skill learning methods (Pertsch et al., 2021; Rosete-Beas et al., 2023), we argue that incorporating an additional posterior distribution to encode the low-level action sequences and learning the latent plan space in a variational way (Kingma, 2013; Van Den Oord et al., 2017) would be beneficial. At the same time, using language instructions as a regularizer might help construct a latent plan space with semantics and interpretability. By decoupling the learning of the high-level policy and low-level policy, we can model the high-level policy as a categorical distribution over the discrete latent plan space, allowing it to capture the multi-modality of the dataset more effectively. For example, given one instruction, there may be multiple potential sub-tasks to choose from next.

In this work, we present **LA**nguage-conditioned **D**iscrete latent plans via semantic **S**kill abstractions (**LADS**) to address the limitation of joint end-to-end training discussed above. Our method consists of three main modules: a high-level policy, a low-level policy, and a trajectory encoder. We use VQ-VAE (Van Den Oord et al., 2017) to jointly learn the low-level policy and the trajectory encoder. This results in a discrete latent plan space, *i.e.*, the VQ-VAE codebook. The high-level policy, conditioned on the language instruction, learns to make predictions in this discrete latent plan space for the next skill to execute. Specifically, it outputs a categorical distribution over the discrete space and is supervised by the latent plan provided by the trajectory encoder over the future trajectory. Therefore, the learning of the high-level policy does not interfere with the latent space. Furthermore, we align the latent plan sequence of each trajectory with its corresponding language instruction to regularize the latent space learned by VQ-VAE. We evaluate LADS in two simulated robotic control environments, LOReL (Nair et al., 2022) and Kitchen (Gupta et al., 2019), both of which have language-conditioned datasets. Our results demonstrate that LADS outperforms state-of-the-art baselines in skill learning and compositional generalization across instructions. Additionally, the ablation study confirms the significance of the proposed modules.

To summarize, our contributions are as follows: (1) We present LADS, a novel hierarchical policy learning framework for skill abstraction from language, decoupling the learning of the language-conditioned high-level policy and the latent plan space. (2) We introduce a trajectory encoder and utilize VQ-VAE to learn a discrete latent space with semantic regularization to guarantee controllability and interpretability. (3) We propose modeling the high-level policy as a categorical distribution to effectively capture the dataset's multi-modality. (4) We demonstrate the superiority of LADS through quantitative comparisons and qualitative latent plan visualizations.

## 2    RELATED WORK

**Hierarchical Policy Learning.** Hierarchical policy learning is a widely explored approach to improve the efficiency and generalization of policies in both RL and IL. Typically, a high-level policy can generate goals as explicit future states (Nair & Finn, 2019; Du et al., 2024; Black et al., 2024), implicit latent plans (Lynch et al., 2020; Pertsch et al., 2021; Rosete-Beas et al., 2023), or language (Hu et al., 2019; Jiang et al., 2019; Chen et al., 2021b). A low-level policy then takes action based on these assigned goals. In RL, the low-level policy is usually learned using information-based objectives (Eysenbach et al., 2018; Laskin et al., 2022; Park et al., 2023) or through joint training with

the high-level policy to maximize environment rewards (Kulkarni et al., 2016; Bacon et al., 2017; Veeriah et al., 2021). In IL, the low-level policy can be trained from an offline dataset using goal-conditioned IL (Kujanpää et al., 2023; Du et al., 2024; Black et al., 2024) or latent variable modeling (Lynch et al., 2020; Pertsch et al., 2021; Rosete-Beas et al., 2023). In this work, we adopt the hierarchical policy learning framework and learn the low-level policy using discrete latent variable modeling (Van Den Oord et al., 2017) from an offline dataset with language instructions.

**Language-Conditioned Policy Learning.** Enabling a policy to follow natural language instruction is a crucial step toward achieving generalist AI. Language can directly serve as a form of task representation for the policy (Hermann et al., 2017; Lynch & Sermanet, 2020; Jang et al., 2022). However, this requires the network to learn the structure of the language space and grounding into the environment from scratch, which presents significant challenges. Recent research uses pretrained large language models (LLMs) or VLMs to provide priors, simplifying language-conditioned training through language grounding (Shridhar et al., 2022; Stone et al., 2023; Gao et al., 2024; Jiang & Lu, 2024) or task decomposition (Huang et al., 2022; Du et al., 2023; Singh et al., 2023). In this work, we train the low-level policy with semantic regularization to ground language with latent plans and thus facilitate the learning of the language-conditioned high-level policy.

**Language-Conditioned Skill Abstractions.** Recent work explores learning semantic and interpretable skills from language-conditioned offline datasets. LISA (Garg et al., 2022) uses an end-to-end hierarchical policy to jointly learn the high-level and low-level policies from the dataset. SkillDiffuser (Liang et al., 2024) improves on this by using a Diffuser (Ajay et al., 2022) as the low-level policy. LCSD (Ju et al., 2024) adopts a one-step hierarchical policy with an auxiliary mutual information objective and a diffusion policy (Ho et al., 2020). LAST (Fu et al., 2024) applies variational temporal inference (Kim et al., 2019) to learn skills but relies on an LLM for segmentation priors, limiting its applicability in environments without language-based action spaces. Our method builds on LISA and improves it by decoupling the learning of latent plans and skills from the learning of the high-level policy, thereby enhancing robustness. Additionally, we introduce a categorical prediction head over the discrete latent plan space for the high-level policy, improving its ability to model the dataset's multi-modality.

## 3 PROBLEM SETUP

We consider a multi-task learning environment modeled as a task-augmented Markov Decision Process (MDP) (Garg et al., 2022). The set of tasks is $\mathcal{T}$, where each task in $\mathcal{T}$ consists of one or more sub-tasks $e \in \mathcal{E}$. That is, $\mathcal{T}$ is a subset of the *powerset* of the sub-task set $\mathcal{E}$, *i.e.*, $\mathcal{T} \subseteq \mathcal{P}(\mathcal{E})$. Each task is described by a natural language instruction $l \in L$, which specifies the sub-tasks included in the task. As shown in Figure 1, language instruction can contain two sub-tasks, *e.g.*, *open drawer* and *turn faucet right.* We assume access to a dataset consisting of $N$ language-conditioned trajectories $\mathcal{D} = \{l^i, s_1^i, a_1^i, \ldots, s_{T_i}^i, a_{T_i}^i\}_{i=1}^N$ collected by a *sub-optimal* policy in the environment, where $s_t \in \mathcal{S}$ denotes the state, $a_t \in \mathcal{A}$ denotes the action, and $T_i$ is the length of the trajectory $i$.

We consider the problem of learning a language-conditioned policy $\pi(a_t|s_t, l)$ that outputs an action $a_t$, given the current state $s_t$ and a language instruction $l$, under the dynamics $P : \mathcal{S} \times \mathcal{A} \to \mathcal{S}$ defined by the task-augmented MDP. The decomposable structure of the task space $\mathcal{T}$ makes this learning problem different from the standard multi-task imitation learning, where each trajectory is independently considered as a single task in a monolithic fashion (Jang et al., 2022; Mees et al., 2022b;a; Black et al., 2024). To improve sample efficiency and generalization, the policy must leverage the shared structure across trajectories, *i.e.*, common sub-tasks, to reduce the task space. However, trajectories are not annotated with the sub-tasks executed at each step. Therefore, the core challenge in this problem setup is to learn and reuse skills for sub-tasks in an unsupervised manner.

## 4 METHOD

In this section, we present the details of our method. We begin by defining an objective for learning the skill-based hierarchical policy and optimizing its lower bound (Section 4.1). We decompose the objective into three components. First, we focus on learning skill abstractions from trajectory via VQ-VAE (Van Den Oord et al., 2017), which provides a discrete latent plan space (Section 4.2). Next, we build the high-level policy as a categorical distribution to predict the discrete latent plan

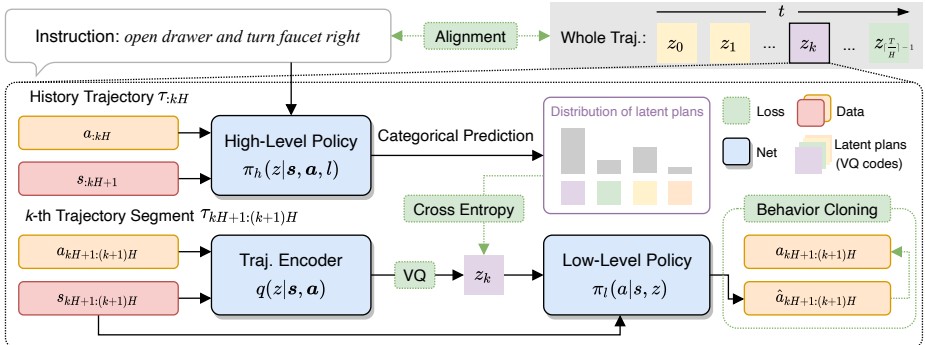

Figure 1: Overview of **LA**nguage-conditioned **D**iscrete latent plans via semantic **S**kill abstractions (**LADS**). For each trajectory segment $\tau_{kH+1:(k+1)H}$, we use a trajectory encoder to map it into a discrete latent plan $z_k$ through vector quantization (VQ). The low-level policy reconstructs actions from $z_k$. Meanwhile, the high-level policy predicts the index of $z_k$ in the discrete latent plan space, based on the history trajectory $\tau_{:kH}$ and language instruction $l$. Lastly, we regularize the latent plan space by aligning the sequence $\{z_0, z_1, \ldots, z_{\lceil \frac{T}{H} \rceil - 1}\}$ of one trajectory to its language instruction.

for the next few steps (Section 4.3). Finally, we impose semantic regularization on the discrete plan space by aligning the sequence of latent plans with language instructions (Section 4.4). We train all modules jointly by combining the proposed losses (Section 4.5).

## 4.1 SKILL-BASED HIERARCHICAL LEARNING

We implements a hierarchical framework consisting of a high-level policy $\pi_h(z|\tau_{:t}, l)$ and a low-level policy $\pi_l(a_t|s_t, z)$. Specifically, the high-level policy takes as input the history trajectory $\tau_{:t} = \{s_1, a_1, \ldots, s_t, a_t, s_{t+1}\}$ and the language instruction $l$, and selects a latent plan $z$ from the latent plan space $\mathcal{Z}$. Once a $z$ is assigned, the low-level policy acts as a skill that executes the latent plan $z$. Following previous work (Garg et al., 2022; Liang et al., 2024), we assume that each skill lasts for $H$ timesteps. We propose the following objective for learning this hierarchical policy,

$$\max_\theta \log p_\theta(\tau_{t+1:t+H}|\tau_{:t}, l), \tag{1}$$

which aims to maximize the likelihood of the future trajectory over the next $H$ timesteps $\tau_{t+1:t+H} = \{s_{t+1}, a_{t+1}, \ldots, s_{t+H}, a_{t+H}\}$ given the history trajectory and the language instruction. $\theta$ denotes the learnable parameters of the hierarchical policy. As for LISA (Garg et al., 2022), its learning objective can be viewed as a lower bound of Equation (1), as detailed in Appendix A.1. However, LISA learns the high-level policy and low-level policy in an end-to-end manner, causing the learning of the language-conditioned policy (high-level policy) to be entangled with the learning of the latent plan space (low-level policy). As a result, LISA has been found to show poor training stability and is prone to cause index collapse in latent space (Ju et al., 2024).

To decouple the learning of the high-level and low-level policies, we introduce a trajectory encoder $q(z|\tau_{t+1:t+H})$, which encodes the ground-truth future trajectory over the next $H$ timesteps into the latent plan space. This allows us to learn the latent plan space in a variational manner, along with the low-level policy. We begin by bounding the learning objective in Equation (1) as follows,

$$\log p(\tau_{t+1:t+H}|\tau_{:t}, l) \geq \mathbb{E}_{q(z|\tau_{:t+H}, l)} \log \frac{p(\tau_{t+1:t+H}, z|\tau_{:t}, l)}{q(z|\tau_{:t+H}, l)}, \tag{2}$$

where $q$ is an approximated posterior. We replace this posterior distribution with our trajectory encoder $q(z|\tau_{t+1:t+H})$, based on the intuition that the single latent plan $z$ should represent the low-level action sequences, relying solely on the future trajectory data. Then we can rewrite the RHS of Equation (2) and get the following learning objective to maximize,

$$\mathcal{J}_{\text{LADS}}(\theta) = \mathbb{E}_{q(z|\tau_{t+1:t+H})} \log \frac{p(\tau_{t+1:t+H}, z|\tau_{:t}, l)}{q(z|\tau_{t+1:t+H})}$$

$$= \mathbb{E}_{q(z|\tau_{t+1:t+H})} \sum_{h=1}^{H} \log p(a_{t+h}|s_{t+h}, z) - D_{\text{KL}}(q(z|\tau_{t+1:t+H}) \| p(z|\tau_{:t}, l)), \tag{3}$$

where constant terms related to environment dynamics are already removed. The detailed derivation process is available in Appendix A.2. By substituting the high-level policy $\pi_h(z|\tau_{:t}, l)$ and low-level policy $\pi_l(a_t|s_t, z)$ into $p(z|\tau_{:t}, l)$ and $p(a_{t+h}|s_{t+h}, z)$ in $\mathcal{J}_{\text{LADS}}(\theta)$, respectively, we obtain the objective for optimizing our skill-based hierarchical framework.

## 4.2 Skill Abstractions

To optimize the first term in $\mathcal{J}_{\text{LADS}}(\theta)$, we implement VQ-VAE (Van Den Oord et al., 2017) to learn the trajectory encoder and low-level policy, resulting in a discrete latent plan space $\mathcal{Z}$. Skills, by design, are often distinct and categorical in nature, such as *open drawer*, *move mug right*, or *pick up kettle*. The discrete latent space provided by VQ-VAE aligns well with this requirement because it forces the model to group similar low-level action sequences into the same cluster. In addition, the discrete latent plan $z$ can enhance the interpretability and controllability of the low-level policy's behavior (Garg et al., 2022; Liang et al., 2024).

Given an input trajectory segment $\tau_{t+1:t+H}$, the trajectory encoder $q(\tau_{t+1:t+H})$[1] maps the segment into a latent vector $\tilde{z}$, which is then quantized to the nearest point in a set of discrete latent codes $\mathcal{Z} = \{z^1, z^2, \ldots, z^M\}$ from the latent codebook of size $M$. This process can be expressed as,

$$z = \arg\min_{z^i \in \mathcal{Z}} \|q(\tau_{t+1:t+H}) - z^i\|^2. \tag{4}$$

The decoder, *i.e.*, the low-level policy $\pi_l(a_t|s_t, z)$, then takes as input this quantized latent vector $z$ and reconstructs the future trajectory $\tau_{t+1:t+H}$, which is used for both the skill execution by the low-level policy and training via the behavior cloning loss,

$$\mathcal{L}_{\text{BC}} = -\sum_{h=1}^{H} \log \pi_l(a_{t+h}|s_{t+h}, z). \tag{5}$$

The VQ-VAE optimization objective includes two parts: the behavior cloning loss for reconstruction and a codebook loss to ensure the discrete latent vectors in $\mathcal{Z}$ are effectively learned,

$$\mathcal{L}_{\text{VQ}} = \mathcal{L}_{\text{BC}} + \|\text{sg}[q(\tau_{t+1:t+H})] - z\|^2 + \beta_{\text{commit}}\|q(\tau_{t+1:t+H}) - \text{sg}[z]\|^2, \tag{6}$$

where $\text{sg}[\cdot]$ denotes the stop-gradient operation, and $\beta_{\text{commit}}$ is a hyperparameter controlling the commitment loss that encourages the encoder to produce $\tilde{z}$ that is close to the quantized vectors.

## 4.3 Discrete Latent Plans

In our learning objective $\mathcal{J}_{\text{LADS}}(\theta)$, the second term is a KL divergence between the trajectory encoder and the high-level policy. This KL loss trains the high-level policy to predict the next latent plan and regularizes the latent plan space. We can treat the two learning processes separately by using a stop-gradient operation (Hafner et al., 2020),

$$\begin{aligned}-D_{\text{KL}}(q(z|\tau_{t+1:t+H})\|p(z|\tau_{:t}, l)) &= -\alpha D_{\text{KL}}(\text{sg}[q(z|\tau_{t+1:t+H})]\|p(z|\tau_{:t}, l)) \\ &\quad - (1-\alpha)D_{\text{KL}}(q(z|\tau_{t+1:t+H})\|\text{sg}[p(z|\tau_{:t}, l)]) \\ &= \alpha\mathcal{J}_p(\theta) + (1-\alpha)\mathcal{J}_q(\theta),\end{aligned} \tag{7}$$

where $\alpha$ controls the balance between the two KL terms. In this section, we describe the design of the high-level policy and its training with the objective $\mathcal{J}_p(\theta)$. The second objective $\mathcal{J}_q(\theta)$ regularizes the latent space and is detailed in Section 4.4.

Given the discrete latent plan space, two approaches can be used to build the high-level policy. The first approach predicts a latent vector $\tilde{z}$ and then quantizes it to the nearest $z$ in the latent codebook, similar to previous methods (Garg et al., 2022; Ju et al., 2024; Liang et al., 2024). This approach trains the high-level policy in a regressive manner, using the latent plan $z$ provided by the trajectory encoder as the target for $\tilde{z}$. The second approach is to predict the index of the latent plan $z$ in the codebook by formulating the high-level policy as a categorical distribution. This allows us to optimize $\mathcal{J}_p(\theta)$ via a cross-entropy loss,

$$\mathcal{L}_{\text{CE}} = -\log \pi_h(\text{id}(z)|\tau_{:t}, l), \tag{8}$$

---

[1]For VQ-VAE, we use a deterministic encoder $q(\tau_{t+1:t+H})$ to replace the distribution $q(z|\tau_{t+1:t+H})$.

where $z$ is selected according to Equation (4) and $\mathrm{id}(z)$ denotes the index of $z$ in the codebook.

We build our high-level policy following the second approach. By predicting discrete latent plans in a categorical way, the high-level policy can more effectively model the dataset's inherent multi-modality. In datasets such as LOReL (Nair et al., 2022), the execution order of sub-tasks in a trajectory does not always follow the annotated language instruction. This means that for a given language instruction, multiple latent plans $z$ could be possible for the following steps. For instance, consider the instruction "*open drawer and turn faucet right*"; Some trajectories might execute sub-task *open drawer* first, while others might execute sub-task *turn faucet right* first. As a result, the skills corresponding to these two sub-tasks may both have high probabilities at the start of the trajectory for this instruction. Furthermore, even trajectories corresponding to the same sub-task can exhibit multi-modality, because the collected data is sub-optimal and might contain multiple modes of behavior, such as performing the sub-task *open drawer* quickly or slowly (Rosete-Beas et al., 2023). Categorical prediction can naturally capture such multi-modal distributions, whereas regressing the value of $z$ may result in sub-optimal predictions between multiple potential options.

## 4.4 SEMANTIC REGULARIZATION

In Equation (7), the objective $\mathcal{J}_q(\theta)$ regularizes the latent plans towards the high-level policy. However, in our method, the high-level policy is modeled as a categorical distribution over the index of the latent plan $z$, while the trajectory encoder directly provides a latent plan $z$. This makes it intractable to compute the KL divergence between them on each trajectory segment. Therefore, we redefine $\mathcal{J}_q(\theta)$ by summing it over the entire trajectory and get,

$$\mathcal{J}_q(\theta) = \log p(l|\boldsymbol{z}), \tag{9}$$

where $\boldsymbol{z} = \{z_0, z_1, \ldots, z_{\lceil \frac{T}{H} \rceil - 1}\}$ represents the sequence of latent plans $z_k$, each encoded from a trajectory segment $\tau_{kH+1:(k+1)H}$ in a trajectory annotated with language instruction $l$. The derivation is detailed in Appendix A.3. $\mathcal{J}_q(\theta)$ regularizes the latent plan space by encouraging the latent plan sequence $\boldsymbol{z}$ of the trajectory to predict its annotated language instruction $l$. This objective does not involve the high-level policy $\pi_h$, so optimizing it naturally stops gradient from backpropagating to the high-level policy and only regularizes the trajectory encoder.

Given the high dimensionality of the language space, it is extremely challenging to train a network that directly predicts the language instruction. Therefore, we implement a surrogate loss function by aligning the sequence of latent plans $\boldsymbol{z}$ with the language instruction $l$, following a contrastive loss approach similar to CLIP (Radford et al., 2021),

$$\mathcal{L}_{\mathrm{align}} = \mathcal{L}_{\mathrm{CL}}(\phi(\boldsymbol{z}), \psi(f_{\mathrm{CLIP}}(l))), \tag{10}$$

where $\phi(\cdot)$ and $\psi(\cdot)$ are two learnable projectors that map $\boldsymbol{z}$ and $l$ into the same dimensional space, respectively, and $f_{\mathrm{CLIP}}(l)$ denotes CLIP text embedding of the instruction $l$. Specifically, $\phi(\boldsymbol{z})$ and $\psi(f_{\mathrm{CLIP}}(l))$ corresponding to the same trajectory are viewed as positive pairs, while other combinations of $\phi(\boldsymbol{z})$ and $\psi(f_{\mathrm{CLIP}}(l))$ are treated as negative pairs. The contrastive loss maximizes the cosine similarity between positive pairs and minimizes the cosine similarity between negative pairs using InfoNCE (Oord et al., 2018). The specific definition of $\mathcal{L}_{\mathrm{CL}}$ is provided in Appendix B.1.

By using this surrogate loss function $\mathcal{L}_{\mathrm{align}}$, we avoid the complexity of directly predicting language instructions. Instead, we impose semantic regularization on the discrete latent space by improving the alignment between latent plans and instructions.

## 4.5 LEARNING LADS

The total loss for LADS is given by combining the losses above,

$$\mathcal{L}_{\mathrm{LADS}} = \mathcal{L}_{\mathrm{VQ}} + \mathcal{L}_{\mathrm{CE}} + \lambda_{\mathrm{align}} \mathcal{L}_{\mathrm{align}}, \tag{11}$$

where $\lambda_{\mathrm{align}}$ controls the weight of the semantic regularization. We optimize the trajectory encoder, high-level policy, and low-level policy jointly. As illustrated in Figure 1, each trajectory is divided into segments $\tau_{kH+1:(k+1)H}$. Among these losses, $\mathcal{L}_{\mathrm{VQ}}$ and $\mathcal{L}_{\mathrm{CE}}$ are calculated on each trajectory segment, while $\mathcal{L}_{\mathrm{align}}$ is calculated over the entire trajectory.

Following previous work (Nair et al., 2022; Garg et al., 2022; Ju et al., 2024; Liang et al., 2024), we use a pretrained DistilBERT (Sanh, 2019) as the language encoder and a causal transformer (Chen

et al., 2021a) as the high-level policy. During the evaluation, the high-level policy provides an index every $H$ timesteps, and the low-level policy takes as input the latent plan $z$ corresponding to this index in the codebook to execute over the next $H$ timesteps. More details on network architecture and training hyperparameters and the pseudocode for training LADS are listed in Appendix B.3.

## 5 EXPERIMENTS

### 5.1 ENVIRONMENTS AND DATASETS

The two environments that we use to evaluate LADS and the corresponding datasets are as follows.

**LOReL** (Nair et al., 2022) is a simulated domain developed on top of Meta-World (Yu et al., 2020), which contains a Sawyer robot interacting with a drawer, a faucet, and two mugs, as shown in Figure 2(a). The robotic arm is controlled using delta end-effector control, and the observation space supports both fully observable **state** observation and partially observable **image** observation. The dataset consists of 50K trajectories, each with 20 timesteps, collected by a random RL policy in the environment and labeled with procedurally generated post-hoc language instructions.

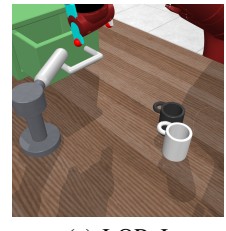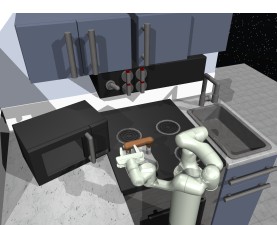

(a) LOReL        (b) Kitchen

Figure 2: Visualization of simulated environments that we evaluate our method on: (a) LOReL, (b) Kitchen.

Following previous work (Nair et al., 2022; Garg et al., 2022; Liang et al., 2024), the evaluation in this environment consists of three parts: (1) **atomic seen** instructions, including 6 tasks, *open drawer*, *close drawer*, *turn faucet left*, *turn faucet right*, *move black mug right*, and *move white mug down*; (2) **atomic rephrased** instructions, containing rephrasings of the atomic seen instructions by replacing the noun or verb, or rewriting the entire instructions; (3) **composite** instructions, consisting of 12 instructions combining two or more atomic instructions, *e.g.*, *open drawer and turn faucet right*. The performance evaluation metric is the success rate. Further details on the LOReL environment settings and evaluation protocols are discussed in Appendix C.1.

**Kitchen** (Gupta et al., 2019) is a simulated domain developed on top of MuJoCo (Todorov et al., 2012), which contains a Franka robot interacting with a kitchen scene including a microwave, a kettle, four oven burners, a light switch, two hinged cabinets, and a sliding cabinet, as shown in Figure 2(b). The robotic arm is controlled by joint position control, and the environment also supports both **state** and **image** observation. The dataset consists of 566 demonstrations collected by humans using VR controllers, with each demonstration manipulating four different elements of the scene in sequence. Since the original dataset lacks language instructions, we procedurally labeled each demonstration based on the four sub-tasks completed in it. For testing, we selected 3 instructions, resulting in a training dataset of 22 instructions and 509 demonstrations. Although the 3 instructions do not appear in the training dataset, the sub-tasks they contain are all present in the training dataset.

The evaluation in this environment includes: (1) **seen** instructions, referring to the 22 instructions in the training dataset, and (2) **unseen** instructions, which are the 3 test instructions not present in the training dataset. We use two performance evaluation metrics: (1) $N$-rate, which counts the number of successfully completed sub-tasks out of four (Gupta et al., 2019; Pertsch et al., 2021); and (2) $K$-rate, which measures the rate at which at least $K$ out of four sub-tasks are completed (Mees et al., 2022b; Shin et al., 2024). More details on dataset construction are provided in Appendix C.2.

### 5.2 BASELINES

We compare LADS with the following baselines for learning language-conditioned skill abstractions. (1) **Lang DT** (Garg et al., 2022): A language-conditioned Decision Transformer (Chen et al., 2021a) as a baseline without hierarchical structure. (2) **LISA** (Garg et al., 2022): A language-conditioned hierarchical learning method that trains the high-level and low-level policies end-to-end, using VQ (Van Den Oord et al., 2017) to discretize the latent space. (3) **LCSD** (Ju et al., 2024):

Table 1: Success rates (%) of different methods on evaluation instructions in LOReL (*upper*) with state observation space and (*lower*) with image observation space. LADS outperforms other methods across most instructions. The best-performing methods are highlighted in **bold**.

| **Instructions** | Lang DT | LISA | LCSD[2] | SkillDiffuser | VAE | Cluster | LADS |
|---|---|---|---|---|---|---|---|
| atomic seen | 27.8±11.5 | 38.6±3.9 | 60.2 | 18.7 | 47.7±6.7 | 32.4±2.2 | **68.3±4.5** |
| atomic rephrased | 21.4±6.9 | 38.4±3.2 | 35.6 | 16.5 | 40.7±4.8 | 26.8±2.6 | **56.9±4.7** |
| composite | 12.5±2.4 | 13.4±4.2 | - | 11.8 | 21.0±3.3 | 11.3±3.1 | **30.0±11.2** |
| atomic seen | 15.0 | 40.0 | 50.8 | 39.3 | 17.4±24.1 | 18.3±10.1 | **52.5±2.1** |
| atomic rephrased | 24.2 | 27.5 | 32.9 | **36.6** | 15.7±22.8 | 11.8±4.8 | 36.2±2.1 |
| composite | 13.3 | 20.9 | - | 20.8 | 8.4±14.2 | 3.7±1.8 | **24.0±0.6** |

Similar to LISA, but adopts a diffusion-based (Ho et al., 2020) low-level policy and a high-level policy that predicts the latent plan at every step, and introduces an additional mutual information objective. (4) **SkillDiffuser** (Liang et al., 2024): Based on LISA, replaces the low-level policy with a Diffuser (Ajay et al., 2022). LCSD and SkillDiffuser are only evaluated in LOReL. (5) **VAE**: Inspired by Pertsch et al. (2021) and Rosete-Beas et al. (2023), this baseline uses VAE (Kingma, 2013) to learn a continuous latent plan space from trajectory segments without semantic regularization. (6) **Cluster**: Inspired by Yuan et al. (2024), this baseline method first uses t-SNE (Van der Maaten & Hinton, 2008) and K-Means (Lloyd, 1982) to acquire a discrete latent plan space from trajectories and then learn the high-level and low-level policies. More detailed descriptions of these methods and their implementations can be found in Appendix D.

## 5.3 EVALUATION RESULTS

Firstly, by evaluating these methods in LOReL, we aim to answer the question: ***How does the performance of LADS in skill learning and composition compare with other baselines?*** Our results in LOReL with state observation and with image observation are shown in Table 1. Success rates are either cited from the original papers or calculated by testing the trained model over 50 episodes. Due to space limitations, a detailed discussion is provided in Appendix D. The results containing standard deviations are calculated from three models trained with different random seeds.

The evaluation results show that LADS outperforms other methods on both atomic seen and composite instructions, demonstrating its superior ability to learn skill abstractions and sequentially combine learned skills. By comparing LADS with LISA, LCSD, and SkillDiffuser, we demonstrate the importance of decoupling the learning of the language-conditioned high-level policy and the latent plan space. Compared to VAE, we confirm the effectiveness of using a discrete latent plan space with semantic regularization. For Cluster in LOReL (image), we use the pretrained ResNet18 (He et al., 2016) to first embed the image observations and then perform clustering and hierarchical learning. However, this approach fails to achieve high success rates. We suspect the reason may be that this method places high demands on good image representation (Yuan et al., 2024), a strong alignment with language in the evaluation environment, such as the use of MineCLIP in Minecraft (Fan et al., 2022). Furthermore, as discussed by Garg et al. (2022), K-Means based on Euclidean distance might not be able to construct an optimal latent space for skill learning. Lastly, the results on atomic rephrased instructions suggest that LADS is robust to variations in language instructions.

Secondly, by evaluating these methods in Kitchen, we aim to answer the question: ***How does the performance of LADS in generalization over unseen combinations of sub-tasks compare with other baselines?*** Results in Kitchen, measured by $N$-rate, are shown in Table 2. $K$-rate are shown in Figure 3 and Appendix E.1. Both metrics are calculated by testing each trained model over 50 episodes, with standard deviations calculated from three models trained with different random seeds.

The evaluation results show that LADS completes the most sub-tasks on both seen and unseen instructions, demonstrating its superior ability to handle long-horizon tasks and generalize to unseen combinations of learned sub-tasks. LISA performs worse than the non-hierarchical baseline, Lang

---

[2]LCSD has neither provided the evaluation result on composite instructions nor open-sourced the code.

Table 2: $N$-rates of different methods on seen and unseen instructions in Kitchen (*upper*) with state observation space and (*lower*) with image observation spapce. LADS outperforms other methods across all instructions. The best-performing methods are highlighted in **bold**.

| Instructions | Lang DT | LISA | VAE | Cluster | LADS |
|---|---|---|---|---|---|
| seen | 1.39±0.12 | 0.71±0.44 | 1.93±0.03 | 0.66±0.09 | **2.25±0.14** |
| unseen | 1.27±0.17 | 0.71±0.53 | 1.79±0.02 | 0.50±0.26 | **2.23±0.16** |
| seen | 1.32±0.41 | 0.93±0.12 | 1.00±0.40 | 0.33±0.08 | **1.61±0.28** |
| unseen | 1.07±0.32 | 1.29±0.20 | 1.27±0.60 | 0.63±0.19 | **1.44±0.32** |

Table 3: Success rates (%) of LADS and its ablation methods on evaluation instructions in LOReL (image). The best-performing methods are highlighted in **bold**.

| Instructions | LADS | w/ VAE | w/o CP | w/o SR |
|---|---|---|---|---|
| atomic seen | **52.5±2.1** | 44.1±7.2 | 43.7±7.8 | 6.2±5.2 |
| atomic rephrased | 36.2±2.1 | **36.6±11.5** | 31.4±5.2 | 5.7±4.6 |
| composite | **24.0±0.6** | 18.2±1.0 | 20.6±3.3 | 1.1±0.3 |

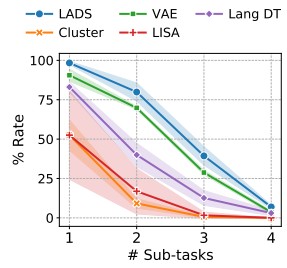

(a) seen instructions

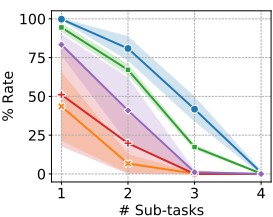

(b) unseen instructions

Figure 3: $K$-rates on (a) seen and (b) unseen instructions in Kitchen (state).

DT, suggesting that the long context increases the learning difficulty for the high-level policy, further complicating the latent space learning. LADS avoids this issue by having the trajectory encoder map much shorter trajectory segments of length $H$ into the latent space, reducing the complexity of learning. Additionally, LADS continues to outperform methods that use a continuous latent space or direct clustering. By $K$-rates shown in Figure 3, we can observe that LADS achieves a higher completion rate for each number of sub-tasks compared to other methods.

## 5.4 ABLATION STUDY

We compare LADS with the following ablation methods to verify the effectiveness of the three components introduced in Section 4. (1) **LADS w/ VAE**: Replaces VQ-VAE with a VAE, resulting in a continuous latent space. Unlike the baseline VAE, this method includes semantic regularization. (2) **LADS w/o CP**: The high-level policy directly predicts the latent plan $z$ instead of predicting a categorical distribution. In this method, the high-level policy is trained using an MSE loss. (3) **LADS w/o SR**: Our method without semantic regularization by setting $\lambda_{\text{align}} = 0$ in $\mathcal{L}_{\text{LADS}}$.

The ablation results in LOReL (image) are shown in Table 3. Replacing VQ-VAE with VAE and using regressive prediction instead of categorical prediction both mildly impair our method, emphasizing the importance of a discrete latent plan space and the ability to model the dataset's multi-modality. Removing semantic regularization from our method significantly affects its performance in LOReL (image). In this environment, aligning the latent plan sequences with instructions helps prevent the discrete latent plan space from overfitting to sub-optimal trajectories by ensuring it encodes semantic information relevant to the sub-tasks. The ablation results in Kitchen can be found in Appendix E.2. We also conduct a hyperparameter study on horizon $H$ and the size of the latent codebook $\mathcal{Z}$ in Appendix E.3.

## 5.5 LATENT PLAN ANALYSIS

We visualize the distributions of latent plans $z$ predicted by the high-level policy learned in LADS to intuitively demonstrate: ***How well does LADS capture multi-modality?*** In LOReL (image), we use composite instructions and their atomic instructions for visualization. For example, one composite instruction "*open drawer and move black mug right*" and the atomic instructions "*open drawer*" and "*move black mug right*" contained in it. We then record the categorical distribution of the discrete

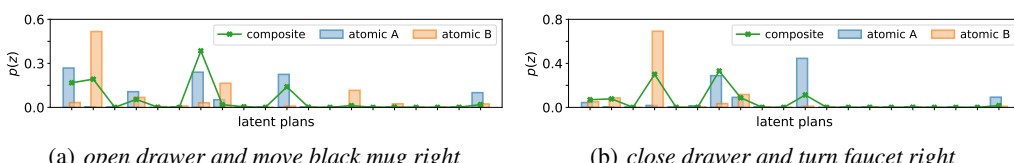

(a) *open drawer and move black mug right*    (b) *close drawer and turn faucet right*

Figure 4: The distributions of latent plans predicted by the high-level policy. The line represents the distribution for the *composite* instruction, while the bars in two colors represent the distributions for two atomic instructions included in the composite instruction, denoted as *atomic A* and *atomic B*.

latent plans predicted by the high-level policy at the start of an episode for each instruction. As shown in Figure 4, the latent plans with high probability for the composite instruction also show high probability for one of its atomic instructions. This means that our high-level policy learns to predict latent plans for a composite instruction by approximately combining possible latent plans for every atomic instruction. Details of visualization process and additional visualization results are available in Appendix E.4.

We demonstrate the interpretability of our discrete latent plans by visualizing the predicted distributions above. We then illustrate the controllable behavior generated by our method. In LOReL (image), we visualize the gripper's trajectory over the first 10 steps of 10 episodes for the composite instruction "*open drawer and move black mug right*" and its atomic instructions. As shown in Figure 5, by comparing the trajectories generated by LADS w/ VAE and LADS, we can observe the advantage of discrete latent plans: it enables the low-level policy to follow one of the

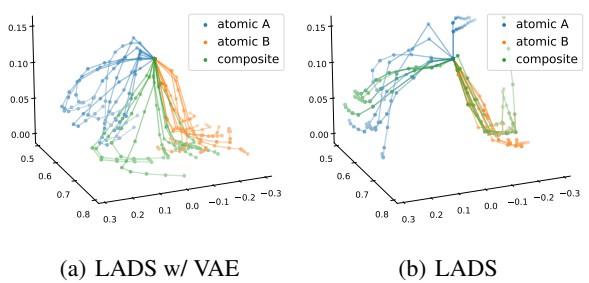

(a) LADS w/ VAE            (b) LADS

Figure 5: Visualization of gripper trajectories in 3D space generated by (a) LADS w/ VAE and (b) LADS for the composite instruction "*open drawer and move black mug right*" and the atomic instructions included in it.

atomic instructions at the beginning, thereby allowing the completion of sub-tasks in sequence. In contrast, a continuous latent plan space causes the low-level policy to generate a trajectory that falls between the two atomic instructions. This occurs because the high-level policy predicts a latent plan $z$ that is between those of the atomic instructions, affecting the model's ability to complete either sub-task. A similar phenomenon is also observed in LADS w/o CP, as discussed in Appendix E.4.

## 6 CONCLUSION

We propose LADS, a novel hierarchical approach for learning skill abstractions from language. Our method employs a discrete latent plan space to learn the low-level policy with semantic regularization and a language-conditioned high-level policy to predict the distribution over discrete plans. Through experiments in two simulated control environments, we demonstrate the effectiveness of LADS and its superiority over language-conditioned hierarchical methods which are trained end-to-end (Garg et al., 2022; Ju et al., 2024; Liang et al., 2024), as well as task-agnostic skill learning approaches (Pertsch et al., 2021; Rosete-Beas et al., 2023; Yuan et al., 2024).

**Limitations and future work**: A limitation of our method is its reliance on a preset horizon $H$, which assumes each skill lasts for a fixed duration. It requires prior knowledge of the dataset and environment to determine a suitable $H$. An alternative approach would be to learn additional termination detection for skills. However, empirical results have shown that this method can easily fall into local optima, treating either a single step or the entire trajectory as a skill (Fu et al., 2024). This approach also requires prior knowledge, such as a minimum horizon for one skill (Kim et al., 2019). Therefore, a key challenge in skill learning domain is how to design a low-level policy that can adapt flexibly to different skill horizons without significantly increasing the learning difficulty at the same time, which is an issue left for future research.

ACKNOWLEDGMENTS

This work was supported by NSFC under Grant 62450001 and 62476008. The authors would like to thank the anonymous reviewers for their valuable comments and advice.

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

## A  LEARNING OBJECTIVE DERIVATION

### A.1  LISA LEARNING OBJECTIVE

LISA (Garg et al., 2022) consists of two modules: a high-level policy $\pi_h(z|\tau_{:t}, l)$ and a low-level policy $\pi_l(a_t|s_t, z)$. LISA optimizes these two modules jointly in an end-to-end manner using behavior cloning loss and VQ loss. The behavior cloning loss is expressed as,

$$\mathcal{L}_{\text{BC}} = -\mathbb{E}_{\pi_h(z|\tau_{:t}, l)} \sum_{h=1}^{H} \log \pi_l(a_{t+h}|s_{t+h}, z). \tag{12}$$

From the learning objective in Equation (1), we have,

$$
\begin{aligned}
\log p(\tau_{t+1:t+H}|\tau_{:t}, l) &= \log \mathbb{E}_{p(z|\tau_{:t}, l)} p(\tau_{t+1:t+H}|z, \tau_{:t}, l) \\
&\geq \mathbb{E}_{p(z|\tau_{:t}, l)} \log p(\tau_{t+1:t+H}|z, \tau_{:t}, l) \\
&= \mathbb{E}_{p(z|\tau_{:t}, l)} \log \prod_{h=1}^{H} p(a_{t+h}|z, \tau_{:t+h-1}, l) p(s_{t+h+1}|s_{t+h}, a_{t+h}) \\
&= \mathbb{E}_{p(z|\tau_{:t}, l)} \sum_{h=1}^{H} \log p(a_{t+h}|z, \tau_{:t+h-1}, l) + C,
\end{aligned}
\tag{13}
$$

where $C$ denotes the constant term related to environment dynamics. By replacing $p(z|\tau_{:t}, l)$ with the high-level policy and $p(a_{t+h}|z, \tau_{:t+h-1}, l)$ with the low-level policy under the assumption that *given the latent $z$, actions taken over the following $H$ timesteps are independent of the past trajectory and language instruction*, we can conclude that minimizing $\mathcal{L}_{\text{BC}}$ is equivalent to maximizing a lower bound of the proposed learning objective in Equation (1).

### A.2  LADS LEARNING OBJECTIVE

We begin by bounding the learning objective in Equation (1) as follows,

$$
\begin{aligned}
\log p(\tau_{t+1:t+H}|\tau_{:t}, l) &= \mathbb{E}_{q(z|\tau_{:t+H}, l)} \log \frac{p(\tau_{t+1:t+H}, z|\tau_{:t}, l)}{p(z|\tau_{:t+H}, l)} \\
&\geq \mathbb{E}_{q(z|\tau_{:t+H}, l)} \log \frac{p(\tau_{t+1:t+H}, z|\tau_{:t}, l)}{q(z|\tau_{:t+H}, l)}.
\end{aligned}
\tag{14}
$$

By replacing $q(z|\tau_{:t+H}, l)$ with the trajectory encoder $q(z|\tau_{t+1:t+H})$, we introduce a heuristic prior into the model that *the single $z$ should represent the low-level action sequences, based solely on the future trajectory data*. Then we can rewrite the lower bound and get the following learning objective,

$$\mathcal{J}(\theta) = \mathbb{E}_{q(z|\tau_{t+1:t+H})} \log \frac{p(\tau_{t+1:t+H}, z|\tau_{:t}, l)}{q(z|\tau_{t+1:t+H})}$$

where $p(\tau_{t+1:t+H}, z|\tau_{:t}, l)$ can be factored as,

$$p(\tau_{t+1:t+H}, z|\tau_{:t}, l) = p(z|\tau_{:t}, l) \prod_{h=1}^{H} p(a_{t+h}|z, \tau_{:t+h-1}, l) p(s_{t+h+1}|s_{t+h}, a_{t+h}). \tag{15}$$

We apply the same assumption as introduced in Appendix A.1 that *given the latent $z$, actions taken over the following $H$ timesteps are independent of the past trajectory and language instruction* by replacing $p(a_{t+h}|z, \tau_{:t+h-1}, l)$ with $p(a_{t+h}|s_{t+h}, z)$. Then,

$$\mathcal{J}(\theta) = \mathbb{E}_{q(z|\tau_{t+1:t+H})} \sum_{h=1}^{H} \log p(a_{t+h}|s_{t+h}, z) - D_{\text{KL}}(q(z|\tau_{t+1:t+H}) \| p(z|\tau_{:t}, l)) + C. \tag{16}$$

### A.3 LADS SEMANTIC REGULARIZATION

We sum the learning objective in Equation (1) over all segments of length $H$ in one trajectory $\tau$ and obtain an equivalent objective as follows,

$$\log p(\tau|l) = \sum_{k=0}^{K} \log p(\tau_{kH+1:(k+1)H}|\tau_{:kH}, l), \tag{17}$$

where $K = \lceil \frac{T}{H} \rceil - 1$ is the number of segments. We can then bound this by,

$$\log p(\tau|l) \geq \mathbb{E}_{q(z|\tau,l)} \log \frac{p(\tau, z|l)}{q(z|\tau, l)}$$
$$= \mathbb{E}_{q(z|\tau,l)} \log \frac{p(\tau|z, l)p(z|l)}{q(z|\tau, l)} \tag{18}$$

Under the same independence assumptions in Section 4.2, Appendix A.1 and Appendix A.2, we replace $q(z|\tau, l)$ with $q(z|\tau)$ and $p(\tau|z, l)$ with $p(\tau|z)$. This allows us to rewrite the lower bound and define the learning objective as,

$$\bar{\mathcal{J}}(\theta) = \mathbb{E}_{q(z|\tau)} \log \frac{p(\tau|z)p(z|l)}{q(z|\tau)}$$
$$= \mathbb{E}_{q(z|\tau)} \log p(\tau|z) - D_{\mathrm{KL}}(q(z|\tau)\|p(z|l)). \tag{19}$$

We factor $q(z|\tau)$ and $p(\tau|z)$ for each trajectory segment $\tau_{kH+1:(k+1)H}$ as follows,

$$q(z|\tau) = \prod_{k=0}^{K} q(z_k|\tau_{kH+1:(k+1)H}), \tag{20}$$

$$p(\tau|z) = \prod_{k=0}^{K} \prod_{h=1}^{H} p(a_{kH+h}|s_{kH+h}, z_k) \cdot C. \tag{21}$$

Then the first term in $\bar{\mathcal{J}}(\theta)$ can be factored as,

$$\mathbb{E}_{q(z|\tau)} \log p(\tau|z) = \sum_{k=0}^{K} \mathbb{E}_{q(z_k|\tau_{kH+1:(k+1)H})} \sum_{h=1}^{H} \log p(a_{kH+h}|s_{kH+h}, z_k) + C, \tag{22}$$

which is equivalent to the summing of first term in $\mathcal{J}_{\mathrm{LADS}}(\theta)$ over the entire trajectory. The second term in $\bar{\mathcal{J}}(\theta)$ regularizes the trajectory encoder at the level of the entire trajectory. Using Bayes' theorem, $p(z|l) = p(l|z)p(z)/p(l)$, we can rewrite this term as follows,

$$-D_{\mathrm{KL}}(q(z|\tau)\|p(z|l)) = \mathbb{E}_{q(z|\tau)} \log \frac{p(l|z)p(z)}{q(z|\tau)p(l)}$$
$$= \mathbb{E}_{q(z|\tau)} \log p(l|z) - D_{\mathrm{KL}}(q(z|\tau)\|p(z)) + C. \tag{23}$$

In particular, the term $-D_{\mathrm{KL}}(q(z|\tau)\|p(z))$ regularizes the latent plans $z$ generated by the trajectory encoder toward a prior distribution $p(z)$. Since we implement the trajectory encoder $q(z|\tau)$ as a VQ, this KL term effectively reduces to the commitment loss that ensures each latent plan $z$ generated by the trajectory encoder is close to the nearest discrete latent vector in the codebook $\mathcal{Z}$. This process encourages the latent plan space to remain well-organized and regularized around a set of discrete latent codes.

Therefore, we can convert the original $\mathcal{J}_q(\theta)$ in Equation (7) from being based on each trajectory segment to being based on the entire trajectory,

$$\mathcal{J}_q(\theta) = \mathbb{E}_{q(z|\tau)} \log p(l|z), \tag{24}$$

which imposes semantic regularization on the discrete latent space by encouraging the sequence of latent plans $z$ over a trajectory to predict the corresponding language instruction. This objective does not involve the high-level policy $\pi_h$, so optimizing it naturally stops gradient from backpropagating to the high-level policy and only regularizes the trajectory encoder.

# B IMPLEMENTATION DETAILS

## B.1 NETWORK ARCHITECTURE

**Language Encoder.** We use a pretrained DistilBERT (Sanh, 2019) as the language encoder, following previous work (Garg et al., 2022; Liang et al., 2024). We freeze its parameters to stabilize the language understanding process.

**Observation Encoder.** For state observations, we use a linear layer to encode the state. For image observations in LOReL (image), we use the convolution layers adopted in previous work (Nair et al., 2022; Garg et al., 2022) to encode the images. In Kitchen (image), we use a pretrained ResNet18 (He et al., 2016) to encode the images, with the ResNet18 parameters kept frozen during training.

**High-Level Policy.** Following LISA (Garg et al., 2022), we use a causal transformer (Chen et al., 2021a) as the language-conditioned high-level policy. The causal transformer takes as input a sequence in the form $\{x_1, x_2, \ldots, x_L, s_1, a_1, s_2, a_2, \ldots, s_t, a_t\}$, where $x_1, x_2, \ldots, x_L$ represent the word embeddings from the language encoder. The output at each state token is passed through a linear layer to produce the logits for the latent plans. The transformer applies a causal mask to hide future trajectory information.

**Low-Level Policy.** For LOReL and Kitchen (state), we use an MLP as our low-level policy. The low-level policy takes the observation and latent plan as input and outputs the predicted action for one step. For LOReL, we implement a 4-layer MLP with Leaky ReLU. For Kitchen (state), we follow Pertsch et al. (2021) and use a 7-layer MLP with Leaky ReLU and 1D batch normalization.

For Kitchen (image), our preliminary experiment results showed that policies without sufficient history information perform poorly in this environment. Therefore, we implement a causal transformer with 2 layers and 4 heads as the low-level policy. It takes as input a sequence in the form $\{z_k, s_{kH-W+1}, a_{kH-W+1}, \ldots, s_{kH}, a_{kH}, s_{kH+1}, a_{kH+1}, \ldots, s_{(k+1)H}, a_{(k+1)H}\}$, which includes the $k$-th trajectory segment, its latent plan $z_k$, and a history window of size $W$. The outputs at state tokens $x_{kH+1}, \ldots, x_{(k+1)H}$ are passed through a linear layer to predict the actions $a_{kH+1}, \ldots, a_{(k+1)H}$. The transformer uses a causal mask to hide future trajectory information. Unlike the MLP-based low-level policy, this history-aware low-level policy leverages more information than the low-level policy used in LISA. So we also conduct an additional experiment to augment the original LISA with our low-level policy in Kitchen (image), and the results are presented in Appendix E.5.

**Trajectory Encoder.** For LOReL and Kitchen (state), we simply implement an LSTM to encode the trajectory segment. It takes as input the concatenation of the encoded state and action at each timestep, where the state is encoded by the observation encoder and the action by a linear layer. The output at the final timestep is passed through a 3-layer MLP with Leaky ReLU to obtain the latent plan $z$, which is then quantized as described in Equation (4). For Kitchen (image), to handle the high dimensionality of ResNet embeddings, we use a 2-layer, 4-head transformer to encode the sequence $s_{kH+1}, a_{kH+1}, \ldots, s_{(k+1)H}, a_{(k+1)H}$ and then take the average over the sequence.

**Semantic Regularization.** We use CLIP ViT-B/32 to encode the instruction into a 512-dimensional embedding. The projector $\psi(\cdot)$ is a 3-layer MLP with ReLU that maps the 512-dimensional language embedding into a 64-dimensional space. Similarly, the projector $\phi(\cdot)$ is a 3-layer MLP with ReLU that maps the $z$ sequence of a trajectory into a 64-dimensional space. For LOReL, where all trajectories have the same length, we concatenate each trajectory's $z$ sequence as the input to $\phi(\cdot)$. For Kitchen, where trajectories vary in length, we introduce an additional transformer to encode the $z$ sequence into an embedding before passing it to $\phi(\cdot)$.

The contrastive loss we implement is as follows,

$$\mathcal{L}_{\mathrm{CL}} = - \sum_{(\boldsymbol{z},l) \sim \mathcal{B}} \log \mathrm{NCE}(\phi(\boldsymbol{z}), \psi(f_{\mathrm{CLIP}}(l))) + \log \mathrm{NCE}(\psi(f_{\mathrm{CLIP}}(l)), \phi(\boldsymbol{z})), \quad (25)$$

where $\mathcal{B}$ represents a batch. $\mathrm{NCE}(\phi(\boldsymbol{z}), \psi(f_{\mathrm{CLIP}}(l)))$ is given by,

$$\mathrm{NCE}(\phi(\boldsymbol{z}), \psi(f_{\mathrm{CLIP}}(l))) = \frac{\exp(\phi(\boldsymbol{z}) \cdot \psi(f_{\mathrm{CLIP}}(l^+))/\tau)}{\sum_{l \in \{l^+, l^-\}} \exp(\phi(\boldsymbol{z}) \cdot \psi(f_{\mathrm{CLIP}}(l))/\tau)}, \quad (26)$$

where $\tau$ is a learnable temperature parameter, $l^+$ is the positive instruction paired with the trajectory, and $\{l^-\}$ are negative instructions paired with other trajectories in the training batch. $\text{NCE}(\psi(f_{\text{CLIP}}(l)), \phi(z))$ is in a similar form by swapping $l$ and $z$ in Equation (26).

## B.2 TRAINING HYPERPARAMETERS

The hyperparameters for the network architecture not covered in Appendix B.1, as well as those related to training, are listed in Table 4. In LOReL, we train each model for 500 epochs. In Kitchen, we train each model for 3000 epochs.

Table 4: Hyperparameters of our experiments.

|  | LOReL (state) | LOReL (image) | Kitchen (state) | Kitchen (image) |
|---|---|---|---|---|
| **High-Level Policy** | | | | |
| hidden size | 128 | 128 | 128 | 128 |
| num layer | 2 | 2 | 4 | 4 |
| num head | 4 | 4 | 4 | 4 |
| dropout | 0.1 | 0.1 | 0.1 | 0.1 |
| **Low-Level Policy** | | | | |
| hidden size | 128 | 128 | 128 | 128 |
| window size $W$ | - | - | - | 10 |
| **Trajectory Encoder** | | | | |
| hidden size | 128 | 128 | 128 | 128 |
| codebook size | 20 | 20 | 20 | 20 |
| codebook dim | 16 | 16 | 16 | 16 |
| $\beta_{\text{commit}}$ | 0.25 | 0.25 | 0.25 | 0.25 |
| VQ EMA update | 0.99 | 0.99 | 0.99 | 0.99 |
| **Semantic Regularization** | | | | |
| hidden size | 256 | 256 | 128 | 128 |
| num layer | - | - | 2 | 2 |
| num head | - | - | 4 | 4 |
| dropout | - | - | 0.1 | 0.1 |
| $\lambda_{\text{align}}$ | 0.0 | 0.4 | 1.0 | 0.01 |
| **Training** | | | | |
| horizon $H$ | 5 | 10 | 20 | 20 |
| batch size | 256 | 256 | 16 | 16 |
| learning rate | 1e-4 | 1e-4 | 1e-4 | 1e-4 |
| optimizer | Adam | Adam | Adam | Adam |

## B.3 ALGORITHM

We provide the pseudocode of LADS as shown in Algorithm 1.

## C ENVIRONMENT AND DATASETS

### C.1 LOREL

**Observation Spaces.** The state observation space for LOReL consists of 15 dimensions, including the poses of robotic joints and objects in the environment. The image observation space for LOReL is a 64×64 RGB image. The dataset contains both state and image observations for each timestep.

---

**Algorithm 1** Training LADS

---

**Require:** Dataset $\mathcal{D}$ consisting of language-paired trajectories, skill horizon $H$, high-level policy $\pi_h$, low-level policy $\pi_l$, trajectory encoder $q$, learnable projectors $\phi$ and $\psi$.
  **while** not converged **do**
    Sample $\tau = (l, \{s_1, a_1, \ldots, s_T, a_T\})$ from dataset $\mathcal{D}$
    **for** $k = 0$ **to** $\lceil \frac{T}{H} \rceil - 1$ **do**
      Set history trajectory $\tau_{:kH} = \{s_1, a_1, \ldots, s_{kH}, a_{kH}, s_{kH+1}\}$
      Set trajectory segment $\tau_{kH+1:(k+1)H} = \{s_{kH+1}, a_{kH+1}, \ldots, s_{(k+1)H}, a_{(k+1)H}\}$
      Acquire a latent code $z_k$ from the trajectory segment $\tau_{kH+1:(k+1)H}$ by Equation (4)
      Compute behavior cloning loss on this segment $\mathcal{L}_{\mathrm{BC}}^k$ by Equation (5)
      Compute VQ-VAE loss on this segment $\mathcal{L}_{\mathrm{VQ}}^k$ by Equation (6)
      Compute cross-entropy loss for the high-level policy on this segment $\mathcal{L}_{\mathrm{CE}}^k$ by Equation (8)
    **end for**
    Obtain $\mathcal{L}_{\mathrm{VQ}}$ by averaging $\left\{\mathcal{L}_{\mathrm{VQ}}^k\right\}$ and obtain $\mathcal{L}_{\mathrm{CE}}$ by averaging $\left\{\mathcal{L}_{\mathrm{CE}}^k\right\}$
    Concatenate the latent codes of all segments $\boldsymbol{z} = \left[z_0, z_1, \ldots, z_{\lceil \frac{T}{H} \rceil - 1}\right]$
    Compute semantic regularization $\mathcal{L}_{\mathrm{align}}$ by Equation (10)
    Update $\pi_h, \pi_l, q, \phi$ and $\psi$ using $\mathcal{L}_{\mathrm{LADS}}$ defined in Equation (11)
  **end while**

---

**Atomic Rephrased Instructions.** We use the same rephrased instructions as in previous work (Garg et al., 2022; Liang et al., 2024), which includes **unseen noun**, **unseen verb**, **unseen noun + verb**, and **human provided** categories. The atomic rephrased instructions consist of 71 distinct variations across all six tasks. We evaluate the performance of each method on these instructions by averaging the results across all individual rephrased instructions, rather than first calculating the average success rate for each category and then averaging across all categories.

**Composite Instructions.** We use the same composite instructions as in previous work (Garg et al., 2022; Liang et al., 2024), which are listed in Table 5.

Table 5: Composite instructions adopted in our evaluation.

| Composite Instructions |
|---|
| open drawer and move black mug right |
| pull the handle and move black mug down |
| move white mug right |
| move black mug down |
| close drawer and turn faucet right |
| close drawer and turn faucet left |
| turn faucet left and move white mug down |
| turn faucet right and close drawer |
| move white mug down and turn faucet left |
| close the drawer, turn the faucet left and move black mug right |
| open drawer and turn faucet counterclockwise |
| slide the drawer closed and then shift white mug down |

### C.2 KITCHEN

**Observation Spaces.** The state observation space for Kitchen consists of 30 dimensions, including the poses of robotic joints and objects in the environment. The image observation space for Kitchen is a 1920×2560 RGB image, which we resize to 224×224 to fit the input requirements

of ResNet18. Since the original dataset only contains state observations, we reconstruct each state using `env.sim.set_state` and render images to obtain the image observations. In our experiments, the image observation space is augmented with a 9-dimensional joint state of the robotic arm. The policy network needs to learn how to extract the poses of task-related objects from the image.

**Language Labeling**. Since the original dataset lacks language instructions, we procedurally labeled each demonstration based on the four sub-tasks completed in it. The atomic instruction corresponding to each sub-task is listed in Table 6. We then generate the instruction for each trajectory by concatenating the four atomic instructions with "and" in order.

Table 6: Atomic instructions labeled to sub-tasks.

| Sub-tasks | Atomic Instructions |
|---|---|
| bottom burner | activate bottom burner |
| top burner | activate top burner |
| light switch | turn on light switch |
| slide cabinet | open sliding cabinet |
| hinge cabinet | open left hinge cabinet |
| microwave | open microwave door |
| kettle | move kettle to top left burner |

**Seen and Unseen Instructions.** We choose the following three instructions for compositional generalization test:

- *Open microwave door and move kettle to top left burner and activate bottom burner and turn on light switch.*
- *Move kettle to top left burner and activate bottom burner and open sliding cabinet and open left hinge cabinet.*
- *Move kettle to top left burner and activate top burner and turn on light switch and open sliding cabinet.*

The difficulty of these instructions increases progressively. For the first instruction, the training dataset includes instructions with the same *first three* sub-tasks. For the second instruction, the dataset has instructions with the same *first two* sub-tasks. However, for the third instruction, the training dataset *does not* include any instruction with the same first two sub-tasks.

In summary, during evaluation, the policy will encounter an unseen sub-task combination after completing three sub-tasks in the first instruction, two sub-tasks in the second instruction, and one sub-task in the third instruction.

## D  BASELINES

**Lang DT.** We directly cite the success rates of Lang DT in LOReL (image) from Garg et al. (2022). Since Garg et al. (2022) does not provide the performance of Lang DT on atomic rephrased instructions and composite instructions in LORel (state), we rerun the code[3] again and report the average performance over three models trained with different random seeds in Table 1. For experiments in Kitchen, we also enlarge the model by double the number of layers in the transformer.

**LISA.** We directly cite the success rates of LISA in LOReL (image) from Garg et al. (2022). Since Garg et al. (2022) does not provide the performance of LISA on atomic rephrased instructions and composite instructions in LORel (state), we rerun the code[3] again and report the average performance over three models trained with different random seeds in Table 1. For experiments in Kitchen, we also enlarge the model by double the number of layers in the transformer.

During our reproduction of LISA in LOReL (state), we encountered the same issue discussed by Ju et al. (2024), where the released code could not achieve normal performance due to index collapse.

---

[3]https://github.com/Div99/LISA

We optimized the code by updating its vector quantization algorithm to the latest version. As shown in Table 1, this partially resolved the index collapse problem, resulting in meaningful success rates. However, we were still unable to reproduce the 66.7% success rate reported in the original paper.

**LCSD.** We directly cite the success rates of LCSD in LOReL (state) and LOReL (image) from Ju et al. (2024).

**SkillDiffuser.** For LOReL (image), we test the trained checkpoint over 50 episodes to obtain a more accurate evaluation, as we found the results to be quite unstable. For LOReL (state), we modify the code[4] to support state observations. This modification does not affect the diffusion model's capability, as it diffuses over the embedding of the state or image rather than the raw observation.

**VAE.** We replace the VQ-VAE module with a VAE while keeping all other settings the same as in our method, except for the semantic regularization. We set the $\beta$ of the KL regularization in the VAE objective to 1e-3 for LOReL and 5e-4 for Kitchen.

**Cluster.** In LOReL (state) and Kitchen (state), we directly apply t-SNE on the states and then use K-Means to obtain 20 clusters, matching the number of codes in our VQ-VAE codebook. In LOReL (image) and Kitchen (image), we apply the same clustering techniques on the ResNet18 embeddings of the images. The low-level policy $\pi_l(a_t|s_t, g)$ is trained by setting the last observation in each trajectory segment as the goal. The high-level policy $\pi_h(\text{id}(g)|\tau_{:t}, l)$ is trained by predicting the index of the cluster center nearest to the last observation in the next trajectory segment.

# E EXTENDED EXPERIMENT RESULTS

## E.1 $K$-RATE IN KITCHEN (IMAGE)

Results in Kitchen (image) measured by $K$-rate are shown in Table 7.

Table 7: $K$-rates of different methods on seen and unseen instructions in Kitchen (image).

| Methods | seen | | | | unseen | | | |
|---|---|---|---|---|---|---|---|---|
| | 1 | 2 | 3 | 4 | 1 | 2 | 3 | 4 |
| Lang DT | 73.9±20.4 | 42.7±14.5 | 12.9±5.1 | 2.0±0.9 | 78.0±12.7 | 25.6±18.0 | 2.9±2.1 | 0.0±0.0 |
| LISA | 62.3±8.6 | 20.8±3.2 | 3.0±0.9 | 0.0±0.0 | 88.9±7.4 | 34.4±11.7 | 5.8±2.1 | 0.2±0.4 |
| VAE | 69.1±20.9 | 26.1±16.4 | 4.1±3.6 | 0.3±0.3 | 76.9±24.3 | 42.9±28.0 | 6.7±9.8 | 0.2±0.4 |
| Cluster | 32.9±8.6 | 0.0±0.0 | 0.0±0.0 | 0.0±0.0 | 63.1±19.3 | 0.0±0.0 | 0.0±0.0 | 0.0±0.0 |
| LADS | 90.6±10.3 | 55.5±17.5 | 14.3±4.9 | 4.1±4.4 | 91.6±7.8 | 47.3±18.3 | 4.9±1.5 | 0.4±0.4 |

## E.2 ABLATION STUDY IN KITCHEN

We also conduct the abation study in Kitchen (state) and the result is shown in Table 8. LADS outperforms all ablation methods on seen and unseen instructions.

Table 8: $N$-rates of LADS and its ablation methods on evaluation instructions in Kitchen (state). The best-performing methods are highlighted in **bold**.

| Instructions | LADS | LADS w/ VAE | LADS w/o CP | LADS w/o SR |
|---|---|---|---|---|
| seen | **2.25±0.14** | 2.01±0.20 | 2.08±0.12 | 2.15±0.03 |
| unseen | **2.23±0.16** | 2.05±0.12 | 1.87±0.11 | 2.04±0.13 |

---

[4]https://github.com/Liang-ZX/SkillDiffuser

### E.3 HYPERPARAMETER STUDY

We analyze the influence of the horizon $H$ and the size of the codebook $\mathcal{Z}$ on the performance of our methods in LOReL (image) and Kitchen (state). As shown in Figure 6(a), overly short horizons can cause the low-level policy to focus on learning primitive action patterns, which is highly noisy, reducing the overall controllability of the hierarchical framework. Conversely, overly long horizons place an excessive burden on the low-level policy by making it learn long-horizon skills.

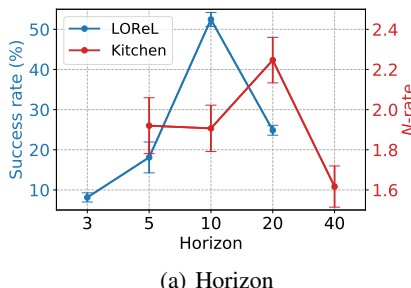
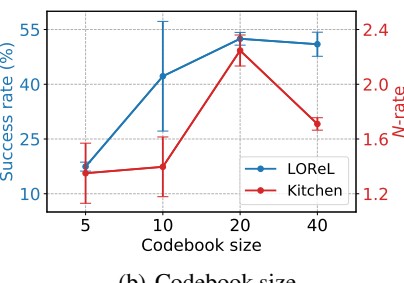

(a) Horizon                         (b) Codebook size

Figure 6: Influence of (a) horizon $H$ and (b) codebook size on success rates in LOReL (image) and $N$-rates in Kitchen (state).

We observe that 20 codes are an appropriate choice for both LOReL and Kitchen for the codebook size, as shown in Figure 6(b). A smaller codebook size is insufficient for the latent space and low-level policy to model the diverse sub-tasks and multi-modality present in the dataset. While a larger codebook size theoretically offers better modeling capacity, it may potentially affect the learning efficiency of the latent space, as observed in Kitchen. In contrast, this influence appears to be minor in LOReL.

### E.4 VISUALIZATION

**Latent plan distribution.** As shown in Figure 4, we demonstrate the distributions of latent plans predicted by the high-level policy. Taking the instruction "*open drawer and move black mug right*" as an example, we illustrate how to obtain the latent plan distributions for both the composite instruction and its corresponding atomic instructions. First, we initialize the LOReL environment and input the composite instruction "*open drawer and move black mug right*" along with the current observation into the high-level policy. The high-level policy outputs the predicted distribution of latent plans, which we visualize as the green lines in Figure 4.

Next, we input the atomic instruction "*open drawer*" and the current observation into the high-level policy to obtain its corresponding latent plan distribution, visualized as the blue bars in Figure 4. Similarly, we input the atomic instruction "*move black mug right*" to obtain its latent plan distribution, which is visualized as the orange bars in Figure 4. We observe that the latent plans with high probabilities for the composite instruction also show high probabilities for one of its atomic instructions.

**Trajectory visualization.** In LOReL, the policy controls a robotic arm. To illustrate the skills learned by LADS, we visualize the trajectory of the end-effector, *i.e.*, the gripper of the robotic arm. As shown in Figure 5, the three axes represent the x, y, and z coordinates of the end-effector. All trajectories start from the same initial position, corresponding to the end-effector's starting point at the beginning of each episode.

**Latent plan visualization.** We visualize the continuous latent plan $z$ predicted by the high-level policy at the start of each trajectory, given the composite instruction: "*open drawer and move black mug right*" and the atomic instructions contained in it. For each instruction, we sample 50 times. We use multidimensional scaling (MDS) to reduce the raw latent plan to 2 dimensions, as this method preserves the distances between data points. As shown in Figure 7(a), the latent plan $z$ output by the high-level policy of LADS w/ VAE for the composite instruction tends to fall between those for the atomic instructions. Consequently, the gripper trajectories generated by LADS w/ VAE exhibit the same pattern, as seen in Figure 5(a). A similar pattern is observed in the latent plan $z$ output by the

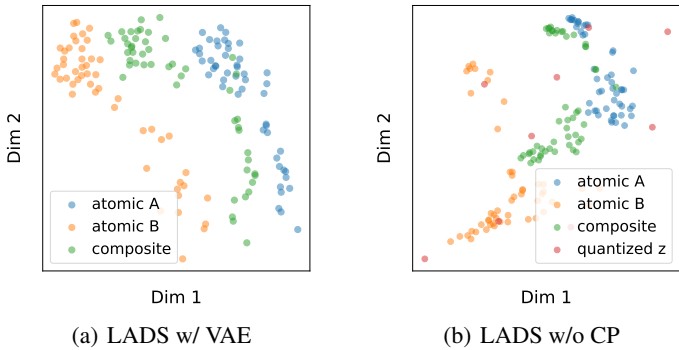

(a) LADS w/ VAE    (b) LADS w/o CP

Figure 7: Continuous latent plan $z$ output by the high-level policy of (a) LADS w/ VAE and (b) LADS w/o CP before quantization, reduced to 2 dimensions.

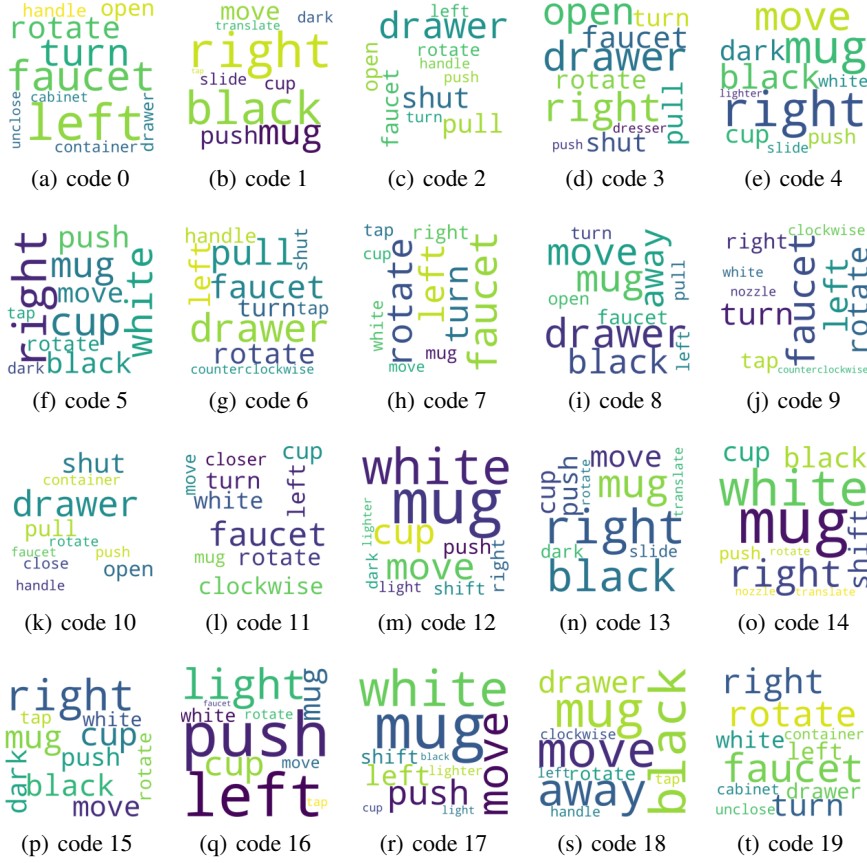

Figure 8: Word clouds in LoReL (image) for each latent plan $z$.

high-level policy of LADS w/o CP. Although the quantization operation ensures that the $z$ provided to the low-level policy is meaningful within the codebook, there is still a potential issue: quantizing $z$ for the composite instruction, which falls between the atomic instruction plans, may result in a sub-optimal $z$ for completing sub-tasks.

**Word clouds.** To enhance the interpretability of our learned latent plan, we present word clouds corresponding to each latent code, ranging from 0 to 19, in LoReL (image). We record the latent plans selected by the high-level policy during the execution of each atomic instruction and then reorganize the data to associate each latent code with the words in its corresponding instructions.

This transformed data is then used to generate word clouds. For each instruction, we execute our learned policy for 50 episodes. As illustrated in Figure 8, each latent code approximately represents a distinct atomic instruction. For example, codes 0, 7, 9, and 11 correspond to "*turn faucet left*", while codes 1, 4, 13, and 15 correspond to "*move black mug right*".

**Latent plan heat map.** Based on the statistical results described above, we also generate heat maps illustrating the relationships between latent plans and text. As shown in Figure 9, the heat maps include: (a) a correlation matrix between each latent plan and individual words; (b) column-wise normalization of this matrix to represent the frequency of each word associated with a given latent plan; (c) row-wise normalization of this matrix to represent the frequency of each latent plan associated with a given word; and (d) a correlation matrix between each latent plan and atomic instructions. The results demonstrate that most of the latent plans learned by LADS are selected during evaluation, and no index collapse, as depicted in Figure 4 (*upper*) of Ju et al. (2024), occurs.

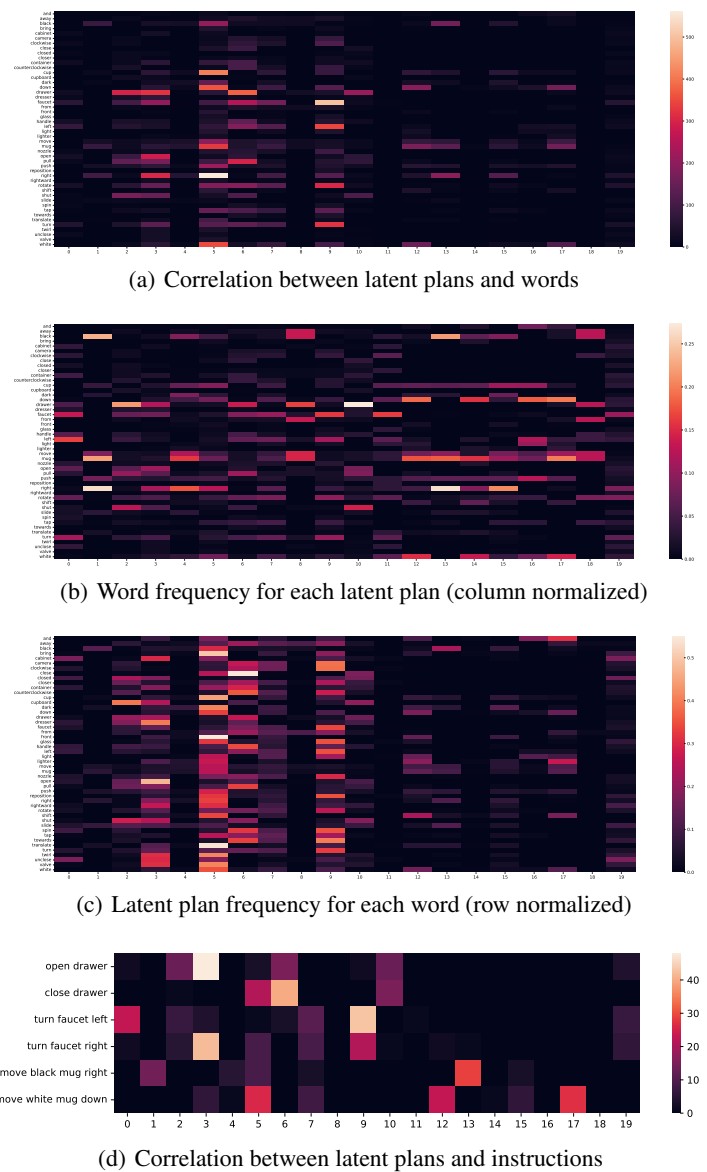

(a) Correlation between latent plans and words

(b) Word frequency for each latent plan (column normalized)

(c) Latent plan frequency for each word (row normalized)

(d) Correlation between latent plans and instructions

Figure 9: Latent plan heat maps.

**Multimodality under the same skill.** As shown in Figure 4 and Figure 9(d), a single atomic instruction or skill can activate multiple latent plans. For instance, in Figure 9(d), the high-level

policy tends to select latent plans 2, 3, 6, and 10 for the instruction "*open drawer*" and latent plans 0 and 9 for "*turn faucet left*". This indicates that multiple latent plans can represent the same skill. To explore this further, we visualize the trajectories generated by manually setting the latent plan for the low-level policy to address the question: ***What distinguishes the behavior patterns of latent plans corresponding to the same skill?*** Similar to Figure 5, we plot the end-effector trajectories over the first 10 steps of 10 episodes for two cases: (1) the low-level policy following latent plans 2, 3, 6, and 10, associated with *open drawer*, and (2) the low-level policy following latent plans 0 and 9, associated with *turn faucet left*, as shown in Figure 10.

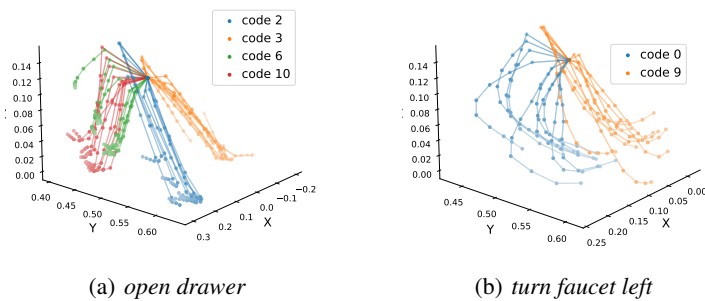

(a) *open drawer*  (b) *turn faucet left*

Figure 10: Visualization of the end-effector's trajectories in 3D space generated by the low-level policy (a) following latent plans 2, 3, 6, and 10, and (b) following latent plans 0 and 9.

As shown in Figure 10(a), the trajectories of latent plans 2, 6, and 10 move in the positive x-axis direction, indicating that the robotic arm is being controlled by the low-level policy to move toward the drawer. In the LOReL environment, the drawer is located along the positive x-axis relative to the robotic arm's starting position. The trajectories of the three latent plans do not fully overlap, suggesting that different latent plans represent *distinct behavioral patterns for accomplishing the same skill*, as discussed in Section 4.3. In this case, they correspond to different spatial paths toward the drawer. In contrast, the trajectory of latent plan 3 moves in the opposite direction, which we speculate represents the sequence of actions involved in pulling the drawer open after the robotic arm has gripped the handle. Similarly, we observe that latent plans 0 and 9 also show different behavior patterns for the same skill *turn faucet left*.

### E.5 LISA WITH OUR LOW-LEVEL POLICY

Table 9: $N$-rates of methods LADS, LISA, and LISA with our low-level policy on evaluation instructions in Kitchen (image).

| Instructions | LADS | LISA | LISA w/ our low-level policy |
|---|---|---|---|
| seen | 1.61±0.28 | 0.93±0.12 | 0.84±0.27 |
| unseen | 1.44±0.32 | 1.29±0.20 | 1.05±0.14 |

In Kitchen (image), we implement a history-aware low-level policy that leverages more information than the low-level policy used in LISA. To ensure a fair comparison, we also augment the original LISA with our history-aware low-level policy. As shown in Table 9, LISA with our low-level policy performs at the same level as the original LISA. Therefore, we can conclude that LISA's failure in Kitchen (image) is due to its end-to-end training approach.

