# OpenReview forum: "Discrete Latent Plans via Semantic Skill Abstractions"
_ICLR.cc/2025/Conference — ICLR 2025 Poster_

### Official Review · Reviewer_N4Ju · 2024-10-29

**Soundness:** 3
**Presentation:** 3
**Contribution:** 3
**Rating:** 8
**Confidence:** 4

**Summary:**

Learning skills from language instructions poses a significant challenge for current algorithms, particularly when dealing with complex tasks that require sequential execution of multiple skills. Previous research (Garg et al. 2022, Liang et al. 2024, Ju et al. 2024) has addressed this issue by unsupervised learning of constituent skills from language-labeled trajectories. Notably, LISA (Garg et al. 2024) has developed a hierarchical approach to learn interpretable skill codes for each task; however, it encounters stability issues. This work improves upon LISA by decoupling the learning of high-level and low-level policies and introducing additional loss terms to enhance stability. Additionally, it addresses dataset multimodality by modeling the high-level policy as a categorical policy. Their approach, **LADS**, demonstrates more stable training, leading to improved generalization and compositionality performance, with ablation studies analyzing the impact of each modification made.

**Strengths:**

* The authors have pinpointed a specific and relevant issue in the prior algorithm, LISA (Garg et al. 2022), and have methodically sought to address it. They hypothesize that the end-to-end training of high-level and low-level policies in LISA leads to instability, and they have developed a new algorithm to overcome this challenge.

* The paper is well-written, with each loss term clearly explained and supported by thorough proofs and mathematical details.

* The authors have conducted extensive experiments, comparing their method to various baselines from previous research, which bolsters the credibility of their approach and hypotheses. The metrics and evaluations align with prior work, facilitating straightforward comparisons.

* They have achieved impressive results in both compositionality and generalization tasks during evaluation.

* The comprehensive ablation studies provide insights into the sources of performance improvements, allowing readers to gain a deeper understanding of the proposed algorithm.

**Weaknesses:**

* The paper argues that instability issues in LISA stem from the end-to-end training of high-level and low-level policies. While LADS decouples these policies, it still trains the trajectory encoder and low-level policy together in an end-to-end manner. How does this approach enhance stability compared to LISA's method? Additionally, the trajectory encoder in LADS has the benefit of looking ahead by $H$ steps before predicting the next skill code $z$. Could this be LADS's primary advantage over LISA?

* The interpretability results presented in Section 5.5 are somewhat unclear and could be better articulated. I have specific questions in the Questions section below, but I am curious about how the skill codes in LADS compare to those in LISA. Are they interpretable? LISA demonstrated a mapping from skill codes to word clouds and behaviors. Do we observe similar behaviors in LADS? Are the skill codes collapsing like in LISA, or do they exhibit more stability?

* The argument for multi-modality is currently presented in a somewhat unconvincing manner. For example, in Section 4.3, the authors state that the dataset may perform the same skill, such as "open drawer," either quickly or slowly, leading to multimodality. How does this impact the categorical representation of skills, considering both variations would still be captured under the same skill? Furthermore, how does the distribution of skills matter if the low-level policy acts on only one skill at a time? Although the ablation results indicate that the categorical distribution performs slightly better, could this be due to the inherent difficulty of predicting the latent code $z$ directly for the high-level policy?

* Semantic regularization appears to play a crucial role in this algorithm, but a compelling rationale for its necessity is lacking. Why do we require both $J_p$ and $J_q$? Why not use just one? What is the significance of the stop gradients? The explanation in lines 474-476 is confusing; it seems that even $J_p$ alone could suffice for language alignment. Why is $J_q$ also needed?

**Questions:**

In addition to the questions mentioned in the Weaknesses section, I have a few specific inquiries about the paper:

* L252: What does it mean when you say the high-level policy regularizes the latent space?

* I find Figure 4 quite confusing. Could you provide a clearer explanation? Specifically, how is the composite distribution calculated? What does the x-axis represent? Why does the probability of the composite distribution approach 0?

* Figure 5 is also somewhat unclear. What are the labels for the axes, and what do the curves represent? Why are we only considering the first 10 steps?

---

> ### Author Response · Authors · 2024-11-22
>
> We appreciate Reviewer N4Ju's efforts and valuable comments. We would like to address the weaknesses and questions raised in the review.
>
> ### **Training stability**
>
> > The paper argues that instability issues in LISA stem from the end-to-end training of high-level and low-level policies. While LADS decouples these policies, it still trains the trajectory encoder and low-level policy together in an end-to-end manner. How does this approach enhance stability compared to LISA's method?
>
> While both LISA and LADS involve joint training of two modules in an end-to-end manner, LADS achieves greater stability and ease of training. This can be explained from two perspectives:
>
> **Reconstruction vs. prediction**: In LADS, the end-to-end learning problem is framed as a **reconstruction task**, where the supervised signal for the low-level policy is inherently present in the input to the trajectory encoder. In contrast, the end-to-end learning problem in LISA is a **prediction task**, where the supervised signal for the low-level policy is not directly contained in the input to the high-level policy. Intuitively, reconstruction is a simpler and more stable learning objective compared to prediction.
>
> **Grounded inputs**: In LADS, the input to the trajectory encoder consists of **states and actions**, which are naturally grounded in the environment. However, in LISA, the input of the high-level policy contains **language embeddings**, which are not grounded in the environment. As discussed in Section 1, this lack of grounding increases the difficulty of the end-to-end learning.
>
> > Additionally, the trajectory encoder in LADS has the benefit of looking ahead by $H$ steps before predicting the next skill code $z$. Could this be LADS's primary advantage over LISA?
>
> We would like to clarify that the trajectory encoder is only used during training, so we address this question separately for training and evaluation.
>
> **During Training**: The trajectory encoder’s ability to look ahead by $H$ steps contributes to the stability of LADS’s training process, as explained above.
>
> **During Evaluation**: In evaluation, the trajectory encoder is not used because the future trajectory for the next $H$ steps is unavailable. LADS uses the same information as LISA, *i.e.*, the history trajectory. This means LADS does not benefit from the trajectory encoder’s lookahead during evaluation. Both methods rely on the high-level policy to predict the next skill code based on the history trajectory and instruction.
>
> Thus, the performance improvement of LADS over LISA is not due to the trajectory encoder’s lookahead but rather stems from the improved training algorithm employed by LADS.
>
> ### **Interpretability**
>
> > I am curious about how the skill codes in LADS compare to those in LISA. Are they interpretable?
>
> We added visualizations of our interpretable latent codes using **word clouds** and **heat maps** in Appendix E.4. These visualizations are generated using the same approach as in LISA.
>
> > LISA demonstrated a mapping from skill codes to word clouds and behaviors. Do we observe similar behaviors in LADS?
>
> From the **word clouds** in Figure 8, it is evident that each latent code corresponds to distinct skills. For instance, codes 0, 7, 9, and 11 correspond to *turn faucet left*, while codes 1, 4, 13, and 15 correspond to *move black mug right*. These results highlight the **interpretability** of the latent codes in LADS.
>
> > Are the skill codes collapsing like in LISA, or do they exhibit more stability?
>
> From the **heat maps** in Figure 9, we observe that most latent plans learned by LADS are utilized during evaluation, with no evidence of index collapse, as was depicted in Figure 4 (upper) of [1]. This demonstrates the **stability** of our method and its ability to avoid the collapse issues observed in LISA.
>
> ### **Multimodality**
>
> > The argument for multi-modality is currently presented in a somewhat unconvincing manner. For example, in Section 4.3, the authors state that the dataset may perform the same skill, such as "open drawer," either quickly or slowly, leading to multimodality. How does this impact the categorical representation of skills, considering both variations would still be captured under the same skill?
>
> To clarify, LADS is designed to learn a distribution over **latent plans**, not directly over skills. This implies that the relationship between latent plans and skills is not one-to-one. As demonstrated in Figures 8 and 9, a single skill can correspond to multiple latent plans. This enables LADS to effectively capture multi-modality within the same skill, such as variations in speed or trajectories. For example, *open drawer quickly* and *open drawer slowly* may be represented by distinct latent plans.

---

> ### Author Response · Authors · 2024-11-22
>
> To verify this, we visualized the end-effector trajectories in 3D space generated by the low-level policy when following different latent plans corresponding to the same skill. The results, presented in the newly added **Figure 10**, show that the trajectories of three latent plans corresponding to the atomic instruction “*open drawer*” do not fully overlap. This indicates that different latent plans can encode **distinct behavioral patterns for achieving the same skill**.
>
> > Furthermore, how does the distribution of skills matter if the low-level policy acts on only one skill at a time?
>
> It is true that the low-level policy only requires one skill at a time for execution, whether the skill is sampled from a distribution or directly generated. However, modeling the multi-modal distribution of skills **during training** is crucial for the high-level policy to function effectively.
>
> Consider scenarios where multiple plausible sub-task sequences exist. For instance, given the instruction “*open drawer and move black mug right*,” both trajectories *open drawer first and then move black mug right* and *move black mug right first and then open drawer* are valid. If the high-level policy does not explicitly model this multi-modal distribution through categorical prediction, it risks **regressing to an averaged or intermediate skill** (*e.g.*, *move towards the space between the drawer and the black mug*) after training. Such a representation is neither interpretable nor actionable, as illustrated in Figure 5(a).
>
> By explicitly modeling the multi-modal distribution with categorical prediction, the high-level policy is able to learn distinct sub-task sequences and select one of the plausible skills during execution. This ensures both interpretability and effectiveness in decision-making.
>
> > Although the ablation results indicate that the categorical distribution performs slightly better, could this be due to the inherent difficulty of predicting the latent code $z$ directly for the high-level policy?
>
> We agree that the inherent difficulty of directly predicting the latent code $z$ may contribute to the performance gap. However, this does not diminish the importance of categorical prediction in **modeling the multi-modality** present in the dataset. To support this, we compare the results of the ablation study on categorical prediction (CP) in the LOReL and Kitchen datasets, as shown in Tables 3 and 8, and summarize them as follows:
>
> |         | LADS | LADS w/o CP | Performance drop (%) |
> | ------- | ---- | ----------- | -------------------- |
> | LOReL   | 52.5 | 43.7        | 16.4                 |
> | Kitchen | 2.25 | 2.08        | 7.56                 |
>
> The results show that removing categorical prediction has a more significant impact on performance in LOReL than in Kitchen. This aligns with the fact that the LOReL dataset exhibits stronger multi-modality compared to the Kitchen dataset. The more multi-modal the dataset, the more critical categorical prediction becomes.
>
> **Why does LOReL dataset exhibit stronger multi-modality**: The LOReL dataset was collected using a random policy, leading to a higher proportion of noisy and diverse trajectories. In contrast, the Kitchen dataset was collected by human demonstrations, resulting in more consistent trajectories with lower variability.
>
> ### **Learning objectives**
>
> > Why do we require both $J_p$ and $J_q$? Why not use just one? What is the significance of the stop gradients?  The explanation in lines 474-476 is confusing; it seems that even $J_p$ alone could suffice for language alignment. Why is also $J_q$ needed?
>
> **Why do we decompose $-D_{\rm KL}(q(z|\tau_{t+1:t+H})\\| p(z|\tau_{:t},l))$ into $J_p$ and $J_q$**: In our method, the high-level policy is modeled as a categorical distribution over the index of the latent plan $z$, while the trajectory encoder directly provides a latent plan $z$. This makes it intractable to compute the KL divergence between them on each trajectory segment. Therefore, we decompose it into $J_p=D_{\rm KL}({\rm sg}[q(z|\tau_{t+1:t+H})]\\| p(z|\tau_{:t},l))$ and $J_q=D_{\rm KL}(q(z|\tau_{t+1:t+H})\\| {\rm sg}[p(z|\tau_{:t},l)])$ to explore alternative optimization strategies.
>
> **What is the significance of the stop gradients**: The stop gradient operators **separate the learning processes** for the high-level policy and the trajectory encoder. Specifically: (1) $J_p$ focuses on learning the high-level policy by preventing gradients from propagating back to the trajectory encoder; (2) $J_q$ regularizes the trajectory encoder by preventing gradients from propagating back to the high-level policy. By separating these two learning processes, we can redefine the individual optimization objectives while preserving equivalence to the original KL divergence. Specifically, $J_p$ is implemented as a cross-entropy loss on each trajectory segment, while $J_q$ is redefined as an alignment objective over the whole trajectory.

---

> ### Author Response · Authors · 2024-11-22
>
> **Can we only use one of $J_p$ and $J_q$**: We cannot use only one of them because both are essential. If we only use $J_p$, we will lose the semantic regularization term, which is proven to be crucial in our experiments. If we only use $J_q$, then the high-level policy would lack an optimization objective.
>
> > L252: What does it mean when you say the high-level policy regularizes the latent space?
>
> The original sentence states: "This KL loss trains the high-level policy to predict the next latent plan and regularizes the latent plan space". The subject of "regularizes the latent space" is actually "This KL loss". Therefore, we intend to convey that it is the KL loss, not the high-level policy, that regularizes the latent space.
>
> ### **Visualization**
>
> > I find Figure 4 quite confusing. Could you provide a clearer explanation? Specifically, how is the composite distribution calculated? What does the x-axis represent? Why does the probability of the composite distribution approach 0?
>
> We newly added detailed explanation on how we acquire the latent plan distributions for composite instructions and atomic instructions in **Appendix E.4 (Latent plan distribution)**. Taking the instruction "*open drawer and move black mug right*" as an example. First, we initialize the LOReL environment and input the composite instruction "*open drawer and move black mug right*" along with the current observation into the high-level policy. The high-level policy outputs the predicted distribution of latent plans, which we visualize as the green lines in Figure 4.
>
> Next, we input the atomic instruction "*open drawer*" and the current observation into the high-level policy to obtain its corresponding latent plan distribution, visualized as the blue bars in Figure 4. Similarly, we input the atomic instruction "*move black mug right*" to obtain its latent plan distribution, which is visualized as the orange bars in Figure 4.
>
> In Figure 4, the x-axis represents the indices of different latent plans (ranging from 0 to 19), and the y-axis represents the probability of each latent plan predicted by the high-level policy. When the probability of certain latent plans approaches 0, it indicates that the high-level policy deems those latent plans irrelevant or mismatched to the given instruction.
>
> > Figure 5 is also somewhat unclear. What are the labels for the axes, and what do the curves represent? Why are we only considering the first 10 steps?
>
> The curves represent the trajectory of the end-effector, *i.e.*, the gripper of the robotic arm, in 3D space. The three axes represent x, y, z coordinates of the end-effector. We added detailed explanations in **Appendix E.4 (Trajectory visualization)**.
>
> We only visualize the first 10 steps for two reasons:
>
> (1) When the instruction is composite, such as "*open drawer and move black mug right*", there are two possible sub-tasks **at the beginning** of the episode: *open drawer* and *move black mug right.* We aim to demonstrate that our high-level policy can clearly select one of these sub-tasks to execute, while LADS w/ VAE fails to do so. Later in the episode, after the first sub-task is completed, only the remaining sub-task is left to execute, eliminating the need to choose between multiple sub-tasks.
>
> (2) Visualizing the entire episode would result in a cluttered figure, reducing clarity.
>
>
>
> [1] Ju et al. (2024). Rethinking Mutual Information for Language Conditioned Skill Discovery on Imitation Learning. In *Proceedings of the International Conference on Automated Planning and Scheduling* (Vol. 34, pp. 301-309).

---

> > ### Comment · Reviewer_N4Ju · 2024-11-23
> > **Thank you for the responses (score increased)**
> >
> > I thank the authors for addressing every single one of my questions and concerns in explicit detail. The additional figures and explanation in the appendix along with the explanations of the loss terms and their requirements in the review have resolved most of my confusions that led to a low score in my initial review. I am convinced with their responses and have increased my score.

---

> > > ### Author Response · Authors · 2024-11-24
> > >
> > > We sincerely thank the reviewer for taking the time to read our rebuttal, and we are delighted to see that the reviewer’s concerns have been resolved.

---

### Official Review · Reviewer_aYjy · 2024-11-01

**Soundness:** 3
**Presentation:** 3
**Contribution:** 3
**Rating:** 6
**Confidence:** 4

**Summary:**

The authors introduce a two stage method which starts by learning a discrete latent space for compressing trajectories and then learns a high-level categorical policy in this latent space. They also add a semantic alignment loss to align latent plans with language instructions. The authors show good results in some language-conditioned BC tasks.

**Strengths:**

Originality:
 * The semantic alignment objective seems to be novel

Quality:
 * The authors ablate the main parts of their method and see that they are helpful
 * The authors achieve strong performance compared to the baselines they include
 * The authors provide a nice experiment showing the multimodal capabilities of their work.

Clarity
 * Overall the paper and its conclusions are easy to follow

Significance
 * Language conditioned imitation learning is an important problem and this paper does a good job trying to address it. I find the insight about the semantic alignment interesting.

**Weaknesses:**

Important issues:
* A key weakness of the work is that it is very similar to several recent works that it does not compare to. In particular, [VQ-BeT](https://sjlee.cc/vq-bet/), [QueST](https://arxiv.org/abs/2407.15840) have very similar setups with a discrete skill learning step followed by a high-level policy learning step. QueST is relatively newer but VQ-Bet is a pretty big oversight, and there should be a comparison to one of or both of these works.
 * VQ-BeT also evaluates in the kitchen environment and it would seem that their results, as well as the results from several baselines, are far better than yours. In particular, you report an average of 2.25 goals achieved while they report 3.78, and their worst-performing non-naive baseline reports 3.09. It's important to add this comparison and explain why that is the case to justify the publication of this paper.

Minor issues:
 * As I mention in the following section, it's difficult to compare the LOREL results to other works because your evaluation schema is different.

**Questions:**

* I find it very surprising that the version of this method without semantic regularization does so poorly. Do you have any insights as to why that might be? Why do other baselines, like VAE, not have a similar problem? Why do algorithms like VQ-BeT, which has a more expressive high-level policy head, not suffer from the same overfitting.
 * Can you please confirm the difference between your work and similar works like VQ-BeT? As far as I can tell, aside from some architectural differences the main difference is the semantic alignment.
 * Line 348 you mention that your evaluation scheme follows previous work, but none of the papers you subsequently cite have the same 'atomic seen / atomic rephrased / composite' division. Can you please explain the relationship between your scheme and the scheme used by LOREL and SkillDiffuser (seen / unseen noun / unseen verb / unseen noun / human provided)? This will help me more directly compare with the results in those works.


Small comments / questions:
 * Equation 10: I don't believe you define $\mathcal{L}_\textrm{CL}$ anywhere

---

> ### Author Response · Authors · 2024-11-22
>
> We appreciate Reviewer aYjy's efforts and valuable comments. We would like to address the weaknesses and questions raised in the review.
>
> ### **Differences with VQ-BeT**
>
> > QueST is relatively newer but VQ-Bet is a pretty big oversight, and there should be a comparison to one of or both of these works.
>
> > Can you please confirm the difference between your work and similar works like VQ-BeT? As far as I can tell, aside from some architectural differences the main difference is the semantic alignment.
>
> We thank Reviewer aYjy for highlighting the related works that also use discrete latent codes for behavior learning. We would like to clarify both the similarities and the essential differences between LADS and VQ-BeT [1].
>
> **Similarities**
>
> Both LADS and VQ-BeT leverage VQ-VAE to learn **discrete latent codes** and a transformer to predict the **categorical distribution** over these discrete codes.
>
> **Differences**
>
> **Abstraction levels**: LADS, following prior works including LISA, LCSD, and SkillDiffuser, focuses on using discrete latent codes for **skill abstraction**, where a skill represents a mapping from states to actions. In contrast, VQ-BeT, along with its predecessors BeT [2] and C-BeT [3], focuses on using discrete latent codes for **action tokenization**, representing action sequences that are independent of states. Skill abstraction operates at a higher level than action tokenization and enables task decomposition. Therefore, latent codes learned by LADS can encapsulate more semantic and interpretable meaning, as shown in Section 5.5 and Appendix E.4. This distinction is also evident in the respective temporal horizons: LADS performs best with horizons greater than 1 (*e.g.*, 10 or 20), while VQ-BeT typically uses action sequence lengths of 1.
>
> **Policy conditioning**: LADS conditions the policy on **language instructions**, whereas VQ-BeT conditions on **goal observation sequences**. As noted in Section 1, goal observations are inherently grounded in the observation space, whereas language-conditioned policies face the additional challenge of grounding language into the observation space. As we will demonstrate in the comparison with VQ-BeT below, directly adopting VQ-BeT results in poor performance in our evaluation environments. In contrast, LADS addresses this challenge by learning discrete latent codes at a higher level and incorporating a **semantic regularization** term.
>
> **Interpretability**: Like prior works including LISA, LCSD, and SkillDiffuser, LADS demonstrates superior interpretability of latent codes. As extensively shown in Section 5.5 and Appendix E.4, the latent codes learned by LADS exhibit strong correlations with atomic instructions. In contrast, VQ-BeT and its predecessors BeT and C-BeT do not demonstrate such interpretability.
>
> In summary, VQ-BeT and LADS follow different directions with distinct focuses. VQ-BeT emphasizes action tokenization for representing low-level action sequences, whereas LADS focuses on skill abstraction for learning high-level, human-interpretable representations.
>
> ### **Comparison with VQ-BeT**
>
> > VQ-BeT also evaluates in the kitchen environment and it would seem that their results, as well as the results from several baselines, are far better than yours. In particular, you report an average of 2.25 goals achieved while they report 3.78, and their worst-performing non-naive baseline reports 3.09. It's important to add this comparison and explain why that is the case to justify the publication of this paper.
>
> We evaluate two methods in Kitchen with state observations:
>
> **L-VQ-BeT**: This method leverages the pretrained VQ-VAE provided by VQ-BeT and uses the same action prediction heads as VQ-BeT. To ensure a fair comparison, all other settings, including the transformer architecture and training paradigm, are kept consistent with LADS.
>
> **LADS w/ VQ-BeT Head**: As discussed earlier, LADS and VQ-BeT operate at different levels of abstraction. To explore potential performance improvement by combining the two methods, we replace the original decoder of LADS with the action prediction head of VQ-BeT. The model is updated by substituting $\mathcal{L}\_{\rm BC}$ in Equation (6) with $\mathcal{L}\_{\rm VQ-BeT}$, scaled by a coefficient of 0.005.
>
> The results are shown in the table below.
>
> |        | Lang DT   | LADS      | L-VQ-BeT  | LADS w/ VQ-BeT Head |
> | ------ | --------- | --------- | --------- | ------------------- |
> | seen   | 1.39±0.12 | 2.25±0.14 | 1.75±0.07 | **2.45±0.13**       |
> | unseen | 1.27±0.17 | 2.23±0.16 | 1.84±0.17 | **2.25±0.10**       |

---

> ### Author Response · Authors · 2024-11-22
>
> **Directly adopting VQ-BeT results in poor performance**. In the original paper, VQ-BeT conditions on goal observation sequences, which are naturally grounded in the observation space. Tasks based on goal observation sequences are easier than the language-conditioned tasks considered in our paper. As noted in Section 1, language-conditioned policies face the additional challenge of grounding language into the observation space. Applying VQ-VAE to low-level actions shows a limited advantage over the naive baseline Lang DT. In contrast, LADS learns VQ-VAE at a higher semantic level with a longer time horizon, easing the learning of the language-conditioned high-level policy.
>
> **LADS and VQ-BeT can be complementary.** Leveraging VQ-BeT’s action prediction head can be seen as providing a more structured action space compared to the original continuous action space. As shown in the table, LADS w/ VQ-BeT Head outperforms standard LADS, consistent with our expectations. This result supports our claim that LADS and VQ-BeT operate at different levels of abstraction and can complement each other.
>
> ### **Semantic Regularization**
>
> > I find it very surprising that the version of this method without semantic regularization does so poorly. Do you have any insights as to why that might be?
>
> LADS w/o SR (without semantic regularization) does not perform poorly in all experimental environments. As shown in Table 8, it achieves 2.15 goals in Kitchen with state observations. We speculate that the significant performance drop observed in LOReL with image observations may be attributed to the following factors:
>
> (1) LOReL dataset contains more noisy actions: The LOReL dataset was collected using a random policy, resulting in a higher proportion of noisy actions. In contrast, the Kitchen dataset was collected through human demonstrations, producing more consistent trajectories with less variability.
>
> (2) Compared to state observations, image observations make the environment partially observable, increasing the learning difficulty for the low-level policy.
>
> These factors likely cause LADS w/o SR to get stuck in some local optima in LOReL with image observations due to the noisy actions.
>
> > Why do other baselines, like VAE, not have a similar problem?
>
> As shown in Table 1 (lower), in LOReL with image observations, VAE also performs poorly, achieving only a 17.4% success rate on atomic seen instructions.
>
> > Why do algorithms like VQ-BeT, which has a more expressive high-level policy head, not suffer from the same overfitting.
>
> Besides the dataset quality discussed above, we believe that leveraging VQ-VAE to cluster the action space can act as a form of regularization, helping to filter out noise and prevent overfitting.
>
> ### **Evaluation metrics**
>
> > Can you please explain the relationship between your scheme and the scheme used by LOREL and SkillDiffuser (seen / unseen noun / unseen verb / unseen noun / human provided)? This will help me more directly compare with the results in those works.
>
> We apologize for the confusion and provide clarification below:
>
> **Atomic seen instructions**: In our paper, these correspond to the "seen" category in previous works.
>
> **Atomic rephrased instructions**: This category includes all "unseen noun", "unseen verb", "unseen noun + verb", and "human provided" instructions from previous works, totaling 71 distinct instructions over six tasks. We evaluate the performance of each method on these instructions by averaging the results across all individual rephrased instructions, rather than first calculating the average success rate for each category and then averaging across all categories. We chose this approach because the four categories contain an uneven number of instructions. We added these details to Appendix C.1.
>
> Further details on how we obtained evaluation results for the baseline methods are provided in Appendix D.
>
> ### **Small comments / questions**
>
> > Equation 10: I don't believe you define $\mathcal{L}\_{\rm CL}$ anywhere
>
> Due to page limitation, the definition of $\mathcal{L}\_{\rm CL}$ was provided in Appendix B.1. To make this clearer, we have now added a reference in Line 309: "The specific definition of $\mathcal{L}\_{\rm CL}$ is provided in Appendix B.1."
>
>
>
> [1] Lee et al. (2024). Behavior Generation with Latent Actions. *Proceedings of the 41st International Conference on Machine Learning*.
>
> [2] Shafiullah et al. (2022). Behavior transformers: Cloning $ k $ modes with one stone. *Advances in neural information processing systems*, *35*, 22955-22968.
>
> [3] Cui et al. (2023). From Play to Policy: Conditional Behavior Generation from Uncurated Robot Data. In *The Eleventh International Conference on Learning Representations*.

---

> > ### Comment · Reviewer_aYjy · 2024-11-22
> >
> > Thanks to the authors for the detailed response. They have helped to clarify why the semantic alignment is so important. A key difference between LADS and comparisons like VQ-BeT is that it operates at a higher level and does a better job mapping between language and the observation space, and the semantic regularization helps with that.
> >
> > My core concern leading to the low score was that I believed the comparison to VQ-BeT was missing and that VQ-BeT outperformed LADS on the same benchmarks. The authors have pointed out that VQ-BeT was operating on a slightly easier version of the benchmark, provided a comparison in their benchmark, and showed that the two methods, which operate at different levels of abstraction, can complement one another.
> >
> > The authors have done a good job addressing my concerns. Considering this and the strengths pointed out by other reviewers I'm happy to increase my score.

---

> > > ### Author Response · Authors · 2024-11-23
> > >
> > > We sincerely thank the reviewer for taking the time to read our rebuttal, and we are delighted to see that the reviewer’s concerns have been resolved.

---

### Official Review · Reviewer_3qsh · 2024-11-02

**Soundness:** 4
**Presentation:** 4
**Contribution:** 2
**Rating:** 8
**Confidence:** 2

**Summary:**

The paper introduces a hierarchical skill learning framework called LADS (LAnguage-conditioned Discrete latent plans via semantic Skill abstractions) aimed at learning skill abstractions from language instructions to enhance skill generalization across diverse tasks. This method is designed to help AI agents interpret and follow human instructions by modeling a discrete latent plan space, which decouples high-level language-based policy from low-level control.
LADS uses a VQ-VAE model to encode sequence of actions into a discrete latent skill space, which allows the high-level policy to predict categorical distributions over potential skills. The hierarchical structure includes three components: a high-level language-conditioned policy, a low-level skill policy, and a trajectory encoder, all of which contribute to stable training. The method demonstrates its effectiveness in robotic control environments like LOReL and Kitchen, outperforming baselines in skill learning and generalization to new task compositions. The key contributions include a decoupled hierarchical policy, improved discrete skill space interpretability, and enhanced multi-modal skill selection.

**Strengths:**

- **Clear Presentation:**
    - The motivation is clearly presented.
    - The related work section is very well-organized, outlining the relationship between this work and others across three different aspects, which helps readers—especially those unfamiliar with the field—understand the context more clearly.
    - The methods are well-decomposed.

- **Significant Performance Improvement on the Benchmark**

**Weaknesses:**

- **Concern about Scalability**

    According to Appendix B.2, the codebook size for the trajectory encoder is 20, which is relatively small. With a larger dataset that includes more diverse atomic instructions, the method would likely require a larger codebook size, increasing the output dimension of the high-level policy. In such cases, I wonder if:

    - *LADS without CP* might outperform, as the high-level policy dimension would not necessarily need to increase.
    - *LADS with VAE* might be advantageous due to its continuous latent space, which doesn’t require an expanded codebook size.
    - A combination of the two, *LADS without CP and with VAE*, might also offer a performance benefit.

- **Concern about Discreteness**

    The set of atomic instructions used in each task appears discrete, meaning each instruction is mutually exclusive. However, if we were to encounter instructions with overlapping meanings but different magnitudes, such as:

    - *open the drawer 5 cm, open the drawer 10 cm, open the drawer 15 cm, ...*
    - *move white mug right 5 cm, move white mug right 10 cm, move white mug right 15 cm, ...*

    In such cases, I’m uncertain that having discrete latent plans would be beneficial for task-solving. Perhaps in these scenarios, *LADS with VAE* (as discussed in Section 5.4), with its continuous latent space, could be more effective. This raises the possibility that the proposed method may be overly tailored to tasks with discrete instructions.

- **Need for Pseudocode**

    Given that the proposed framework involves training multiple components, with each component requiring several neural networks, including pseudocode would provide readers with a clearer, holistic understanding of the process.

- **Minor Correction**

    - *Line 192:* The objective function does not include the parameter θ. It may be clearer to adjust the notation to explicitly include θ in the objective function.

**Questions:**

Could you respond to the following two concerns I raised in the weaknesses section?
- *Concern about Scalability*
- *Concern about Discreteness*

---

> ### Author Response · Authors · 2024-11-22
>
> We appreciate Reviewer 3qsh’s efforts and valuable comments. We would like to address the weaknesses and questions raised in the review.
>
> ### **Concern about scalability**
>
> > According to Appendix B.2, the codebook size for the trajectory encoder is 20, which is relatively small. With a larger dataset that includes more diverse atomic instructions, the method would likely require a larger codebook size, increasing the output dimension of the high-level policy. In such cases, I wonder if:
>
> We agree that experiments with a larger, more diverse dataset would be ideal. However, to the best of our knowledge, no such datasets are currently available. To address this scalability concern, we draw insights from research involving discrete representations, particularly in the field of computer vision.
>
> > *LADS without CP* might outperform, as the high-level policy dimension would not necessarily need to increase.
>
> We agree that LADS w/o CP (without categorical prediction) may offer advantages in certain aspects, such as avoiding increased output dimensions of the high-level policy when the codebook size expands. However, LADS w/o CP models the high-level policy as $S \times L \to Z$, directly predicting a single latent plan $z$.  **This approach limits its ability to capture the underlying multi-modal and complex distributions of the next latent plans**. In contrast, LADS models $S \times L \to \Delta(Z)$, a distribution over the latent space. As datasets and codebook sizes grow, LADS w/o CP’s inability to model distributions becomes increasingly problematic, leading to degraded performance.
>
> Furthermore, there is supporting evidence from related fields. In computer vision tasks such as image generation, autoregressive methods predict a categorical distribution over the next token rather than directly generating a deterministic token. These methods (*e.g.*, DALLE [1] and VQ-GAN [2]) use codebooks much larger than that in LADS (*e.g.*, size 8192) while achieving strong performance. This evidence demonstrates that increasing the codebook size and the output dimension of the high-level policy is feasible and effective.
>
> **Beyond categorical prediction:** If categorical prediction is not used, alternative methods for predicting latent plans, such as **diffusion models**, could be employed. This is because diffusion models have the capacity to fit complex distributions. However, such approaches introduce significant additional complexity and fall outside the scope of this paper, leaving them as a direction for future research.
>
> > *LADS with VAE* might be advantageous due to its continuous latent space, which doesn’t require an expanded codebook size.
>
> Like LADS w/o CP, LADS w/ VAE models the high-level policy as $S \times L \to Z$, lacking the ability to model distributions over latent plans. As datasets becomes larger and more diverse, this limitation will lead to degraded performance.
>
> Meanwhile, LADS’s performance is unlikely to be significantly impacted by expanded codebook sizes. Research on VQ-VAE demonstrates robust scalability with codebook size up to 2^11 [3]. Larger codebooks allow finer-grained representations without degrading performance or interpretability.
>
> > A combination of the two, *LADS without CP and with VAE*, might also offer a performance benefit.
>
> LADS w/ VAE inherently does not use categorical prediction, so a combination of the two is equivalent to LADS w/ VAE. As mentioned earlier, this approach lacks the ability to model multi-modal distributions effectively, which remains a key limitation.
>
> ### **Concern about discreteness**
>
> > However, if we were to encounter instructions with overlapping meanings but different magnitudes, such as:
> >
> > - *open the drawer 5 cm, open the drawer 10 cm, open the drawer 15 cm, ...*
> > - *move white mug right 5 cm, move white mug right 10 cm, move white mug right 15 cm, ...*
> >
> > In such cases, I’m uncertain that having discrete latent plans would be beneficial for task-solving.
>
> Firstly, we would like to address the question: **can discrete latent plans represent skills with overlapping meanings but different magnitudes?**
>
> **Theoretical perspective**: If the **codebook size is sufficiently large**, discrete representations can be extended to encode skills with overlapping semantics but varying magnitudes. For example, *open drawer by 5 cm*, *10 cm*, and *15 cm* can be assigned separate codes in the latent space. This retains the interpretability of discrete representations while enabling fine-grained control over tasks.

---

> ### Author Response · Authors · 2024-11-22
>
> **Empirical observations**: In our experiments, we observed that multiple latent plans can represent the same atomic instruction, resulting in trajectories that differ in execution but achieve the same semantic goal. These observations, now included in **Appendix E.4 (Multimodality under the same skill)**, demonstrate that discrete representations can effectively handle multi-modality and subtle differences in execution, even within skills sharing overlapping semantics.
>
> Although we did not explicitly observe latent plans representing variations in magnitudes in the current experiments, the observed patterns suggest that such phenomena might emerge as the dataset and codebook size scale up. Exploring this phenomenon is a promising avenue for future work, when more diverse and suitable datasets become available. Currently available language-paired datasets do not support such fine-grained instructions.
>
> Therefore, we argue that discrete latent representations possess the capability to represent skills with overlapping meanings but different magnitudes.
>
> > Perhaps in these scenarios, *LADS with VAE* (as discussed in Section 5.4), with its continuous latent space, could be more effective.
>
> Based on the discussion above, both LADS and LADS w/ VAE are theoretically capable of modeling skills with overlapping meanings and different magnitudes. However, as highlighted in our paper, **LADS w/ VAE has a significant limitation**: its inability to effectively learn the dataset’s multi-modality.
>
> For instance, consider the instruction "*open drawer 10 cm*". Both LADS and LADS w/ VAE can select a latent plan corresponding to this skill. However, for composite instructions like "*open drawer 10 cm and move white mug right 5 cm*", LADS w/ VAE may encounter the issue shown in Figure 5(a), generating a trajectory that falls between *open drawer* and *move white mug right* due to the continuous latent space. In contrast, LADS would avoid this problem by predicting a categorical distribution where latent plans for *open drawer 10 cm* and *move white mug right 5 cm* have high probabilities. LADS can then select one of these latent plans, ensuring the trajectory aligns with one sub-task rather than falling between the two.
>
> Finally, we propose potential extensions of our method to improve its capability of representation when instructions include fine-grained control such as magnitude. One promising approach is to use a **multi-level VQ-VAE** [4]. The first level captures high-level semantic distinctions (e.g., *open drawer* vs. *move mug*), while the second level encodes finer details like magnitudes (*5 cm*, *10 cm*, *15 cm*). Alternatively, a **hybrid representation** could combine discrete and continuous latent spaces. The discrete code could represent *open drawer*, while a continuous latent variable specifies the fine-grained instruction (e.g., *5 cm* vs. *10 cm*).
>
> ### **Need for pseudocode**
>
> We added pseudocode in Appendix B.3 to provide readers with a clearer understanding of our algorithm.
>
> ### **Minor correction**
>
> > *Line 192:* The objective function does not include the parameter $\theta$. It may be clearer to adjust the notation to explicitly include $\theta$ in the objective function.
>
> We added a subscript $\theta$ to $p$ in Equation (1) and included additional clarification on line 194: "$\theta$ denotes
> the learnable parameters of the hierarchical policy".
>
>
>
> [1] Esser et al. (2021). Taming transformers for high-resolution image synthesis. In *Proceedings of the IEEE/CVF conference on computer vision and pattern recognition* (pp. 12873-12883).
>
> [2] Ramesh et al. (2021). Zero-shot text-to-image generation. In *International conference on machine learning* (pp. 8821-8831). Pmlr.
>
> [3] Mentzer et al. (2024) Finite Scalar Quantization: VQ-VAE Made Simple. In *The Twelfth International Conference on Learning Representations*.
>
> [4] Zeghidour et al. (2021). Soundstream: An end-to-end neural audio codec. *IEEE/ACM Transactions on Audio, Speech, and Language Processing*, *30*, 495-507.

---

> ### Comment · Reviewer_3qsh · 2024-11-28
>
> Thank you for addressing my questions. I have no further inquiries and am happy to stick with my original score!

---

> > ### Author Response · Authors · 2024-11-29
> >
> > We sincerely thank the reviewer for taking the time to read our rebuttal, and we are delighted to see that the reviewer’s questions have been addressed.

---

### Official Review · Reviewer_Rvu8 · 2024-11-03

**Soundness:** 3
**Presentation:** 4
**Contribution:** 3
**Rating:** 6
**Confidence:** 4

**Summary:**

This paper introduces a method for for language-guided. Hierarchical RL. The paper learns a high-level and low-level policy connected through a latent plan space. Unlike previous approaches, these policies are not learned end-to-end but are decoupled by learning a discrete latent space for the high-level policy to sample from. The paper shows results on two language-instructed control environments.

**Strengths:**

The paper is extremely well presented and written. The methodology is quite complex, but the good use of figures (Fig. 1) and a clearly written methodology section (that explains the motivation behind each step of deriving the losses rather than just throwing math at you) makes the method easy to follow.

The paper is clear on what it sets out to do: learn a better way of jointly learning HRL policies in a latent planning space while avoiding some of the mode collapse and other issues of previous works and this goal is validated through experiments.

The experimental evidence of the claims of the paper appear to be well backed up in two environments. There are a few possibly concerns in the experiments, or ways to maybe make this stronger (see weaknesses), but holistically the experiments do show the methodology is effective, which is good.

**Weaknesses:**

The related work on language-guided HRL is a little bit sparse and skews very recent. Some early important works here such as [1], [2], [3] should be included and also possibly serve as inspiration for more simple baselines (see below) for language-based HRL.

I do wonder if there are some simpler, language based HRL baselines that could have been compared to. Language DT is a good baseline, so appreciate that, but I wonder if there is a simple hierarchical baselines that could be done here as well. From the above works, people had much simpler methods of avoiding the collapse between the hierarchical policies. For instance, in [2], the high-level policy was learned through IL and then held fixed while the low-level policies were fine-tuned, with the intermediate space being a hidden state, rather than in language space.

There are quite a few loss functions that go into this method, L_VQ, the VQ-VQA objective, L_BC the behavioral cloning objective, L_CE, the categorical objective and L_align, the alignment objective. I was really hoping to see a more thorough ablation of all of these losses, but did not see them. Some of these, (w/o VAE or CP) sort of get at these, but not entirely. I would have in particular been curious on how important the BC loss was to these results (it's kind of a footnote in the description of the method but I suspect it's rather important to getting it to work), and how important some of these other losses are, the alignment loss in particular. Especially because this work is quite complicated and incorporates so many losses, it seems important to know how vital each of them actually is (even if for some it will quite obviously fail without, that's good to know).

[1] Yiding Jiang, Shixiang Shane Gu, Kevin P Murphy, and Chelsea Finn. Language as an abstraction for hierarchical deep reinforcement learning. In Advances in Neural Information Processing
Systems, pp. 9414–9426, 2019.
[2] Chen, Valerie, Abhinav Gupta, and Kenneth Marino. "Ask Your Humans: Using Human Instructions to Improve Generalization in Reinforcement Learning." International Conference on Learning Representations 2021.
[3] Hengyuan Hu, Denis Yarats, Qucheng Gong, Yuandong Tian, and Mike Lewis. Hierarchical decision making by generating and following natural language instructions. In Advances in neural
information processing systems, pp. 10025–10034, 2019.

**Questions:**

Line 362ish
What does "procedurally label[ing]" the demonstrations mean? Like, use the simulator to manually label what each of the trajectories are? This isn't really explained.

---

> ### Author Response · Authors · 2024-11-22
>
> We appreciate Reviewer Rvu8’s efforts and valuable comments. We would like to address the weaknesses and questions raised in the review.
>
> ### **Simpler language-based HRL**
>
> > The related work on language-guided HRL is a little bit sparse and skews very recent. Some early important works here such as [1], [2], [3] should be included and also possibly serve as inspiration for more simple baselines (see below) for language-based HRL.
>
> We appreciate Reviewer Rvu8 for highlighting these language-based HRL works \[1\]\[2\]\[3\]. While these works and LADS fall under the intersection of language and HRL, there are notable differences:
>
> 1. These works \[1\]\[2\]\[3\] use natural language to connect the high-level policy and the low-level policy, whereas LADS uses latent variables with a much smaller space. Specifically, denoting the hierarchical policy $\pi(a|s)=\pi_h(g|s)\pi_l(a|s,g)$, where $\pi_h$ and $\pi_l$ represents the high-level and low-level policies, respectively. Then, $g$ in \[1\]\[2\]\[3\] is natural language, while in LADS, it is a latent variable.
> 2. These works \[1\]\[2\]\[3\] leverage **grounded language supervision** during training. For instance, \[2\]\[3\] use supervised learning for training the high-level policy, and \[1\] requires a mapping function from states to natural language. In contrast, LADS does not make such assumptions and learns the latent representation between the high-level and low-level policies in an **unsupervised** manner.
>
> As a result, these simpler baselines rely on assumptions beyond the problem setting considered in this paper, making them unsuitable for comparison in our experiments.
>
> ### **Thorough ablation of losses**
>
> > I was really hoping to see a more thorough ablation of all of these losses, but did not see them.
>
> As shown in Equation (6) and (11), our loss functions include: $\mathcal{L}_{\rm BC}$, $\mathcal{L}_{\rm VQ}$, $\mathcal{L}_{\rm CP}$, $\mathcal{L}_{\rm align}$. The ablation study results for the latter three have already been presented in Section 5.4: LADS w/ VAE, LADS w/o CP, and LADS w/o SR. **Note that w/o SR refers to "without semantic regularization", which corresponds to the ablation for $\mathcal{L}_{\rm align}$.** In addition, by ablation study for $\mathcal{L}_{\rm VQ}$, we mean replacing VQ-VAE with VAE, rather than removing this term from the total loss.
>
> Following the reviewer's suggestion, we added an ablation study for $\mathcal{L}\_{\rm BC}$. As shown in Equation (6), $\mathcal{L}\_{\rm BC}$ is included within $\mathcal{L}\_{\rm VQ}$ and is used to train the low-level policy and trajectory encoder. First, we retain $\mathcal{L}\_{\rm BC}$ but **stop its gradients from back-propagating to the trajectory encoder**, denoted as SG. In this setup, $\mathcal{L}\_{\rm BC}$ is used solely to update the low-level policy, while the trajectory encoder is updated using $\mathcal{L}\_{\rm align}$ and the commitment loss from the VQ-VAE. Next, we **remove $\mathcal{L}\_{\rm BC}$ entirely**. In this case, the low-level policy will not update due to the absence of a loss function, remaining as a random policy.
>
> We summarize the results already presented in the paper and the newly added results, as shown in the table below. We report the success rates on atomic seen instructions in LOReL and N-rates on seen instructions in Kitchen. Given that the ablation results for $\mathcal{L}\_{\rm VQ}$, $\mathcal{L}\_{\rm CP}$ and $\mathcal{L}\_{\rm align}$ have already been discussed in Section 5.4, we focus here on the ablation results for the two variations of $\mathcal{L}\_{\rm BC}$ mentioned above.
>
> | $\mathcal{L}_{\rm BC}$ | $\mathcal{L}_{\rm VQ}$ | $\mathcal{L}_{\rm CP}$ | $\mathcal{L}_{\rm align}$ | LOReL    | Kitchen   |
> | ---------------------- | ---------------------- | ---------------------- | ------------------------- | -------- | --------- |
> | ✔︎                      | ✔︎                      | ✔︎                      | ✔︎                         | 52.5±2.1 | 2.25±0.14 |
> | ✔︎                      | ✔︎                      | ✔                      |                           | 6.2±5.2  | 2.15±0.03 |
> | ✔︎                      | ✔︎                      |                        | ✔︎                         | 43.7±7.8 | 2.08±0.12 |
> | ✔︎                      |                        | ✔︎                      | ✔︎                         | 44.1±7.2 | 2.01±0.20 |
> | SG                     | ✔︎                      | ✔︎                      | ✔︎                         | 51.7±5.0 | 0.88±0.09 |
> |                        | ✔︎                      | ✔︎                      | ✔︎                         | 0.0±0.0  | 0.0±0.0   |

---

> ### Author Response · Authors · 2024-11-22
>
> Stopping the gradients of $\mathcal{L}\_{\rm BC}$ from propagating back to the trajectory encoder has a minor impact on LADS in LOReL but a more significant impact in Kitchen. We hypothesize that this is because trajectories in the LOReL dataset are much shorter (20 steps), allowing the trajectory encoder to learn a meaningful latent plan space with $\mathcal{L}\_{\rm align}$ alone. Short latent plan sequences would be easier to align with the language instructions. In contrast, trajectories in the Kitchen dataset are much longer (ranging from 181 to 409 steps), resulting in much longer latent plan sequences. Under these conditions, the trajectory encoder struggles to learn effectively when relying solely on $\mathcal{L}\_{\rm align}$.
>
> Completely removing $\mathcal{L}\_{\rm BC}$ makes the algorithm unable to achieve meaningful performance, as the high-level policy cannot accomplish tasks by relying on a randomly behaving low-level policy.
>
> ### **Language labeling**
>
> > Line 362ish What does "procedurally label[ing]" the demonstrations mean? Like, use the simulator to manually label what each of the trajectories are? This isn't really explained.
>
> Due to space constraints, we describe the process of procedurally labeling each demonstration in Appendix C.2. Specifically, we labeled each demonstration based on the four sub-tasks completed in it. We assigned an atomic instruction to each sub-task. For example, the sub-task *bottom burner* corresponds to "activate bottom burner" and *microwave* corresponds to "open microwave door", as detailed in Table 6. We then generated the instruction for each trajectory by concatenating the four atomic instructions in order with "and".
>
> [1] Jiang et al. (2019). Language as an abstraction for hierarchical deep reinforcement learning. *Advances in Neural Information Processing Systems*, *32*.
>
> [2] Chen et al. (2021). Ask Your Humans: Using Human Instructions to Improve Generalization in Reinforcement Learning. In *International Conference on Learning Representations*.
>
> [3] Hu et al. (2019). Hierarchical decision making by generating and following natural language instructions. *Advances in neural information processing systems*, *32*.

---

> > ### Comment · Reviewer_Rvu8 · 2024-11-25
> >
> > So is the claim here is that unlike the earlier works, this work does not require sub-episode language instructions correct? So the comparisons would not be fair (these works have more information). I think that's fair as far as direct comparison of numbers goes, but I do still think that this distinction from prior work should still at least be discussed.
> >
> > Thank you for the additional ablation experiments, I think that's super helpful
> >
> > Thank you for the clarification on procedural labeling.

---

> > > ### Author Response · Authors · 2024-11-26
> > >
> > > We sincerely thank the reviewer for taking the time to read our rebuttal, and we are delighted to see that the reviewer’s concerns have been resolved. We are happy to address any further questions.
> > >
> > > > So is the claim here is that unlike the earlier works, this work does not require sub-episode language instructions correct?
> > >
> > > Yes, that is correct. LADS follows research on language-conditioned skill abstractions (LISA, LCSD, SkillDiffuser), which assumes that language instructions are provided at the level of the entire episode, rather than sub-episodes.
> > >
> > > > I think that's fair as far as direct comparison of numbers goes, but I do still think that this distinction from prior work should still at least be discussed.
> > >
> > > We agree with the reviewer’s suggestion. The referenced works use language as an intermediary representation for the hierarchical policy, which should be included under *Hierarchical Policy Learning* in Section 2. In response, we have updated the text by explicitly including *language* as one of the goal types generated by the high-level policy (see Lines 102–103). Additionally, we have added **Appendix F. Extended Related Work** to provide a detailed discussion on the distinctions between these works and LADS.

---

### Meta-Review · Area_Chair_1Lc7 · 2024-12-21

**Metareview:**

This paper introduces a method for for language-guided. Hierarchical RL. The paper learns a high-level and low-level policy connected through a latent plan space. Unlike previous approaches, these policies are not learned end-to-end but are decoupled by learning a discrete latent space for the high-level policy to sample from. The paper shows results on two language-instructed control environments.

Overall, the paper seemed to provide a clear and novel approach that significantly outperformed baselines. In terms of weaknesses, the overall method design is the paper is a bit complex, and upon close reading of the paper, some choices seemed a bit arbitrary. For instance, the authors use CLIP for semantic alignment due to the claim that language is very high dimensional, but many of the language instructions in the considered environments are fairly simple. In addition, the experimental evaluation is on existing environments where performance gains are marginal. To further strengthen the paper, the authors could show some novel capabilities the approach enables.

**Additional Comments On Reviewer Discussion:**

The reviewers and authors had a productive conversation where issues about scalability, baselines and clarifications were made.

---

### Decision · Program_Chairs · 2025-01-22

Accept (Poster)